# Verifier-Constrained Flow Expansion for Discovery Beyond the Data

**Riccardo De Santi**[12][*], **Kimon Protopapas**[1][*], **Ya-Ping Hsieh**[1], **Andreas Krause**[12]
[1]ETH Zürich, [2]ETH AI Center
{rdesanti,kprotopapas,krausea}@ethz.ch, {yaping.hsieh}@inf.ethz.ch

## Abstract

Flow and diffusion models are typically pre-trained on limited available data (e.g., molecular samples), covering only a fraction of the valid design space (e.g., the full molecular space). As a consequence, they tend to generate samples from only a narrow portion of the feasible domain. This is a fundamental limitation for scientific discovery applications, where one typically aims to sample valid designs beyond the available data distribution. To this end, we address the challenge of leveraging access to a verifier (e.g., an atomic bonds checker), to adapt a pre-trained flow model so that its induced density expands beyond regions of high data availability, while preserving samples validity. We introduce formal notions of *strong* and *weak verifiers* and propose algorithmic frameworks for *global* and *local flow expansion* via probability-space optimization. Then, we present **F**low **E**xpander (FE), a scalable mirror descent scheme that provably tackles both problems by verifier-constrained entropy maximization over the flow process noised state space. Next, we provide a thorough theoretical analysis of the proposed method, and state convergence guarantees under both idealized and general assumptions. Ultimately, we empirically evaluate our method on both illustrative, yet visually interpretable settings, and on a molecular design task showcasing the ability of FE to expand a pre-trained flow model increasing conformer diversity while preserving validity.

## 1 Introduction

Recent years have seen major progress in large-scale generative modeling. In particular, flow (Lipman et al., 2022; 2024) and diffusion models (Sohl-Dickstein et al., 2015; Song & Ermon, 2019; Ho et al., 2020) now produce high-fidelity samples and have been applied successfully across domains such as chemistry (Hoogeboom et al., 2022), biology (Corso et al., 2022), and robotics (Chi et al., 2023). These models are typically trained via divergence minimization objectives, such as score (Song et al., 2020) or flow matching (Lipman et al., 2022), to approximate the distribution induced by training data (e.g., molecular samples), which typically only cover a tiny subset of the full valid design space. As a consequence, pre-trained generative models concentrate their density over a narrow region of valid designs, and are unlikely to generate valid samples beyond areas of high data availability. This is a fundamental limitation for scientific

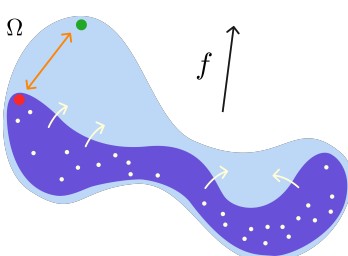

Figure 1: Limited coverage of the valid design space leads to generating sub-optimal samples for downstream optimization tasks.

discovery tasks such as material design and drug discovery, where one typically wishes to generate valid designs beyond the data distribution. In particular, limited coverage of the valid design space leads to an irreducible sub-optimality gap in *generative optimization* (De Santi et al., 2025b; Uehara et al., 2024b; Li et al., 2024) problems, where one aims to generate samples maximizing a task-specific utility function $f : \mathcal{X} \to \mathbb{R}$ (e.g., binding affinity for protein docking), as illustrated in Figure 1.

Prior work has addressed this issue through manifold-exploration schemes that re-balance a pre-trained model's density over diverse, promising modes (e.g., De Santi et al., 2025a;b; Celik et al., 2025). However, the validity signal learned by a pre-trained flow model diminishes outside high-data

---

[*]Equal contribution. Corresponding author: rdesanti@ethz.ch.

regions. Therefore, seemingly promising low-probability modes that such methods would further explore, could turn out to be invalid. This highlights the need to inject further validity information into the exploration process via an external *verifier* (Botta et al., 2025; Wang et al., 2024): formally, a function $v : \mathcal{X} \to \{0, 1\}$ that provides data-specific validity signal. Luckily, there exists more-or-less accurate verifiers for a wide variety of real-world discovery applications, such as atomic-bond checkers for drug discovery (e.g., O'Boyle et al., 2011), protein folding predictors for protein design (e.g., Jumper et al., 2021), as well as physics-based simulators for mechanical and material design (e.g., Kresse & Furthmüller, 1996). Motivated by these insights, in this work we advance flow and diffusion-based design space exploration methods (De Santi et al., 2025a;b) by asking the following question:

*How can we leverage a given verifier to adapt a flow or diffusion model to generate designs beyond high data-availability regions while preserving validity?*

Answering this would contribute to the algorithmic-theoretical foundations of *generative exploration*, and enable applications of flow-based exploration schemes in diverse scientific discovery tasks.

**Our approach** We address this challenge by formally introducing two verifier types. A *strong verifier* is a function $v : \mathcal{X} \to \{0, 1\}$ that characterizes validity exactly (i.e., $v(x) = 1$ iff $x$ is valid). A *weak verifier* instead acts as a *filter*: it rejects certain invalid designs but misses others (formally $v(x) = 0 \implies x$ is invalid). While the former is arguably rare in scientific discovery applications, the latter is ubiquitous. For instance, most molecular checkers examine specific constraints (e.g., atomic bonds, graph topology, or conformer geometry), ruling out certain invalid samples, but without guaranteeing validity. We show that strong verifiers allow to adapt a pre-trained model to *globally* expand over the entire valid design space. While this is not the case for weak verifiers, they can also be leveraged for a more conservative, *local* expansion. To this end, we introduce mathematical frameworks for *global* and *local flow expansion* via verifier-constrained entropy maximization (Sec. 3). Next, we propose **F**low **E**xpander (FE), a scalable mirror descent scheme acting over the flow process noised state space that provably tackles both problems by sequentially alternating expansion and projection steps (Sec. 4). We provide theoretical guarantees for FE, showing convergence results under both idealized and general assumptions via mirror-flow theory (Sec. 5). Ultimately, we evaluate our method on both illustrative, yet visually interpretable settings, and on a molecular design task, showcasing the ability of FE to expand a pre-trained flow model to increase molecular conformer diversity while better preserving validity than current flow-based exploration schemes (Sec. 6).

**Our contributions** In this work, we provide the following contributions:

- A formalization of *Global* and *Local Flow Expansion* via verifier-constrained entropy maximization, which formally capture the practically relevant problem of expanding the coverage of a pre-trained flow or diffusion model by integrating information from an available strong or weak verifier (Sec. 3).
- *Flow Expander (*FE*)*, a principled probability-space optimization scheme that provably solves both problems introduced via constrained entropy maximization over the flow noised state space (Sec. 4).
- **N**oised **S**pace **E**xploration (NSE), a noised state space unconstrained exploration scheme, obtained as a by product, that outperforms existing flow-based methods for high-dim. exploration (Sec. 4).
- A theoretical analysis of the proposed algorithm providing convergence guarantees under both simplified and realistic assumptions via mirror-flow theory (Sec. 5).
- An experimental evaluation of FE, showcasing its practical relevance on both visually interpretable illustrative settings, and on a molecular design task aiming to increase conformer diversity. (Sec. 6).

## 2 BACKGROUND AND NOTATION

**Mathematical Notation.** Using $\mathcal{X} \subseteq \mathbb{R}^d$ to refer to the design space (an arbitrary set), we denote the set of Borel probability measures on $\mathcal{X}$ with $\boldsymbol{P}(\mathcal{X})$, and the set of functionals over the set of probability measures $\boldsymbol{P}(\mathcal{X})$ as $\boldsymbol{F}(\mathcal{X})$. Given an integer $N$, we define $[N] := \{1, \dots, N\}$.

**Flow-based Generative Modeling.** Generative models aim to approximately replicate and sample from a data distribution $p_{data}$. Flow models tackle this problem by modeling a *flow*, which incrementally transforms samples $X_0 = x_0$ from a source distribution $p_0$ into samples $X_1 = x_1$ from the target distribution $p_{data}$ (Lipman et al., 2024; Farebrother et al., 2025). Formally, a *flow* is a time-dependent map $\psi : [0, 1] \times \mathbb{R}^d \to \mathbb{R}$ such that $\psi : (t, x) \to \psi_t(x)$. A *generative flow model* is then defined as a continuous-time Markov process $\{X_t\}_{0 \le t \le 1}$ generated by applying a flow $\psi_t$ to $X_0$, i.e. $X_t = \psi_t(X_0)$, $t \in [0, 1]$ such that $X_1 = \psi_1(X_0) \sim p_{data}$. In the context of flow modeling, the flow $\psi$ is defined by a *velocity field* $u : [0, 1] \times \mathbb{R}^d \to \mathbb{R}^d$, which is a vector field implicitly defining

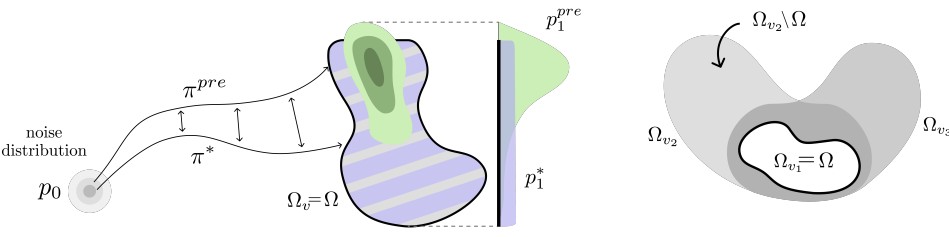

(a) Global flow expansion        (b) Strong and weak verifier sets

Figure 2: (2a) Pre-trained and globally expanded flow model inducing densities $p_1^{pre}$ and optimal density $p_1^*$. (2b) Valid design space $\Omega$, strong and weak verifiers $\Omega_{v_i}$, $i \in [3]$, and their compositions.

$\psi$ via the following ordinary differential equation (ODE), typically referred to as *flow ODE*:

$$\frac{\mathrm{d}}{\mathrm{d}t}\psi_t(x) = u_t(\psi_t(x)), \quad \psi_0(x) = 0 \tag{1}$$

We write $\{p_t\}_{t\in[0,1]}$ to refer to the probability path of *marginal densities* of the flow model, i.e., $X_t = \psi_t(X_0) \sim p_t$, and denote by $p^u$ the probability path of marginal densities induced by the velocity field $u$. In practice, Flow Matching (FM) (Lipman et al., 2024) can be used to estimate a velocity field $u^\theta$ s.t. the probability path $p^{u_\theta}$ satisfies $p_0^{u_\theta} = p_0$ and $p_1^{u_\theta} = p_{data}$, where $p_0$ denotes the source distribution, and $p_{data}$ the target data distribution. Typically FM is rendered tractable by defining $p_t^u$ as the marginal of a conditional density $p_t^u(\cdot|x_0, x_1)$, e.g.,:

$$X_t \mid X_0, X_1 = \kappa_t X_0 + \omega_t X_1 \tag{2}$$

where $\kappa_0 = \omega_1 = 1$ and $\kappa_1 = \omega_0 = 0$ (e.g., $\kappa_t = 1 - t$ and $\omega_t = t$). Then $u^\theta$ can be learned by regressing onto the conditional velocity field $u(\cdot|x_1)$ (Lipman et al., 2022). Interestingly, diffusion models (Song & Ermon, 2019) (DMs) admit an equivalent ODE-based formulation with identical marginal densities to their original SDE dynamics (Lipman et al., 2024, Chapter 10). Consequently, while in this work we adopt the notation of flow models, our contributions carry over directly to DMs.

**Continuous-time Reinforcement Learning.** We formulate continuous-time reinforcement learning (RL) as a specific class of finite-horizon optimal control problems (Wang et al., 2020; Jia & Zhou, 2022; Treven et al., 2023; Zhao et al., 2024). Given a state space $\mathcal{X}$ and an action space $\mathcal{A}$, we consider the transition dynamics governed by the following ODE:

$$\frac{\mathrm{d}}{\mathrm{d}t}\psi_t(x) = a_t(\psi_t(x)) \tag{3}$$

where $a_t \in \mathcal{A}$ is a selected action. We consider a state space $\mathcal{X} := \mathbb{R}^d \times [0, 1]$, and denote by (Markovian) deterministic policy a function $\pi_t(X_t) := \pi(X_t, t) \in \mathcal{A}$ mapping a state $(x, t) \in \mathcal{X}$ to an action $a \in \mathcal{A}$ such that $a_t = \pi(X_t, t)$, and denote with $p_t^\pi$ the marginal density at time $t$ induced by policy $\pi$. Considering the continuous-time reinforcement learning formulation above, a velocity field $u^{pre}$ can be interpreted as an action process $a_t^{pre} := u^{pre}(X_t, t)$, where $a_t^{pre}$ is determined by a continuous-time RL policy via $a_t^{pre} = \pi^{pre}(X_t, t)$ (De Santi et al., 2025a). Therefore, we can express the flow ODE induced by a pre-trained flow model by replacing $a_t$ with $a^{pre}$ in Eq. 3, and denote the pre-trained model by its policy $\pi^{pre}$, which induces a density $p_1^{pre} := p_1^{\pi^{pre}}$ approximating $p_{data}$.

## 3 PROBLEM STATEMENT: GLOBAL AND LOCAL FLOW EXPANSION

Given a pre-trained flow model $\pi^{pre}$ inducing a density $p_1^{pre}$ that covers sufficiently only a limited region of the *valid design space*[1] $\Omega \subseteq \mathbb{R}^d$ (e.g., a molecular space, see Fig. 1), we aim to adapt it by leveraging a verifier (e.g., an atomic bonds checker) to compute a model $\pi^*$, inducing a process:

$$\frac{\mathrm{d}}{\mathrm{d}t}\psi_t(x) = a_t^*(\psi_t(x)) \quad \text{with} \quad a_t^* = \pi^*(x, t), \tag{4}$$

such that its density $p_1^* := p_1^{\pi^*}$ is more uniformly distributed over the valid design space than $p_1^{pre}$. To this end, we first denote by *verifier* a scalar function $v : \mathcal{X} \to \{0, 1\}$, and indicate by $\Omega_v = \{x \in \mathcal{X} : v(x) = 1\}$ the *verifier-set* induced by $v$. Next, we classify any verifier $v$ as *strong* or *weak* depending on the relationship between its verifier-set $\Omega_v$ and the valid design space $\Omega$.

---

[1]In this work, we consider the valid design space to be an unknown, yet bounded set.

**Definition 1** (Strong Verifier). *We denote by* strong verifier *a function* $v : \mathcal{X} \to \{0, 1\}$ *s.t.* $\Omega_v = \Omega$.

By Def. 1, $v(x) = 1 \iff x \in \Omega$, hence a strong verifier fully characterizes the valid design space $\Omega$.

### 3.1 An Idealized Problem: Global Flow Expansion via Strong Verifiers

Given a pre-trained flow model $\pi^{pre}$ and a strong verifier $v : \mathcal{X} \to \{0, 1\}$ as defined within Def. 1, one can capture the problem of computing a new flow model $\pi^*$ such that its marginal density $p_1^* := p_1^{\pi^*}$ covers $\Omega$ uniformly via the following verifier-constrained entropy maximization problem.

---
**Global Flow Expansion via Verifier-Constrained Entropy Maximization**

$$\pi^* \in \underset{\pi : p_0^* = p_0^{pre}}{\arg\max} \ \mathcal{H}(p_1^\pi) \quad \text{subject to} \quad s.t. \quad p_1^\pi \in \boldsymbol{P}(\Omega_v) \tag{5}$$
---

In this formulation, the constraint $p_0^\pi = p_0^{pre}$ enforces that the marginal density at $t = 0$ matches the pre-trained model marginal, and $\mathcal{H} \in \boldsymbol{F}(\Omega_v)$ denotes the differential entropy functional expressed as:

$$\mathcal{H}(\mu) = - \int \mathrm{d}\mu \log \frac{\mathrm{d}\mu}{dx}, \quad \mu \in \boldsymbol{P}(\Omega_v) \tag{6}$$

where $\Omega_v$ is the bounded verifier-set induced by $v$. Crucially, Problem 5 computes a flow model $\pi^*$ inducing the density $p_1^{\pi^*}$ with maximum entropy among all densities supported on $\Omega_v$. Therefore, the optimal density $p_1^{\pi^*}$ according to Problem 5 corresponds to the uniform density over the entire valid design space $\Omega$, i.e., $p_1^{\pi^*} = \mathcal{U}(\Omega)$ - as uniforms are the entropy-maximizing densities on bounded sets and $\Omega_v = \Omega$ due to $v$ being a strong verifier. Notably, Problem 5 does not depend on the prior generative model $\pi^{pre}$. In fact, since the strong verifier $v$ fully characterizes the valid design space $\Omega$, prior information is not required to compute the maximally explorative, yet valid flow model $\pi^*$.

Problem 5 provides a sharp data-free objective for verifier-based flow/diffusion model learning, well capturing the ideal goal of a uniform prior over the valid design space for subsequent use in downstream tasks. Nonetheless, as discussed in Sec. 1, strong verifiers are arguably rare in most scientific discovery applications (e.g., material design, drug discovery). Towards overcoming such limitation, in the following we sharpen our notion of verifier to one that is ubiquitous in real-world discovery tasks.

### 3.2 A Realistic Framework: Local Flow Expansion via Weak Verifiers

We first relax the notion of strong verifier introduced in Def. 1 to the following one of *weak verifier*.

**Definition 2** (Weak Verifier). *We denote by* weak verifier *a function* $v : \mathcal{X} \to \{0, 1\}$ *s.t.* $\Omega_v \supset \Omega$.

As Fig. 2b illustrates, Def. 2 requires only the one-sided condition $v(x) = 0 \implies x \notin \Omega$; unlike strong verifiers, it does not guarantee $v(x) = 1 \implies x \in \Omega$ (i.e., $v$ cannot certify membership in $\Omega$). Instead, it represents a superset $\Omega_v \supset \Omega$ and effectively acts as a *filter*. Moreover, multiple weak verifiers $\{v_i\}$ can be combined, yielding $\Omega_v = \bigcap_i \Omega_{v_i}$, which is typically tighter to $\Omega$ for more diverse verifiers $v_i$, e.g., checking atomic bonds, molecular graph topology, and conformer geometry.

Given this new realistic notion of verifier, the global flow expansion Problem 5 would evidently no longer compute the desired flow model. In fact, for a weak verifier $v$ it holds $\Omega_v \supset \Omega$, therefore the optimal flow density $p^* = \mathcal{U}(\Omega_v)$ would generate invalid designs over $\Omega_v \setminus \Omega$, as shown in Fig. 2b. Moreover, weak verifiers typically induce unbounded verifier sets, which would even render Problem 5 ill-posed. To address these issues, we introduce the *local flow expansion* problem, which aims to locally expand the prior flow model $\pi^{pre}$ by integrating information from both $v$ and $\pi^{pre}$.

---
**Local Flow Expansion via KL-regularized Verifier-Constrained Entropy Maximization**

$$\pi^* \in \underset{\pi : p_0^* = p_0^{pre}}{\arg\max} \ \mathcal{H}(p_1^\pi) - \alpha \mathcal{D}_{KL}(p_1^\pi \| p_1^{pre}) \quad \text{subject to} \quad s.t. \quad p_1^\pi \in \boldsymbol{P}(\Omega_v) \tag{7}$$
---

Here, the weak verifier $v$ acts as a filter preventing the entropy term from driving exploration into verifier-rejected regions. Since $v$ cannot detect all invalid areas, expansion must remain conservative and leverage the validity signal encoded in the prior model. This is achieved via the $\alpha$-weighted KL divergence between the density $p_1^\pi$ induced by the fine-tuned model, and $p_1^{pre}$. Crucially, this term enforces $\pi^*$ to preserve prior validity signal, thus preventing $\pi^*$ from allocating density in regions unlikely according to $\pi^{pre}$, even if valid according to the weak verifier. For sufficiently large $\alpha$, the

density induced by the expanded flow model $\pi^*$ stays arbitrarily close to the prior in probability space - hence *local* expansion. In practice, the choice of $\alpha$ should reflect the degree of risk-aversion versus novelty-seeking toward the discovery task at hand, as well as the quality of the weak verifier $v$ (i.e., how tightly $\Omega_v$ approximates $\Omega$). Interestingly, in the limit of $\Omega_v \to \Omega$, $\alpha$ should clearly be set to 0, which naturally retrieves the presented global flow expansion Problem 5 as a sub-case.

## 4 FLOW-EXPANDER : SCALABLE GLOBAL AND LOCAL EXPANSION

In the following, we propose **F**low **E**xpander (FE), which provably solves the global and local flow expansion problems (see Eq. 5 and 7). To this end, we first lift their formulations from the probability space associated to the last time-step marginal $p_1^\pi$ to the entire flow process $\mathbf{Q}^\pi = \{p_t^\pi\}_{t \in [0,1]}$.

**Flow Expansion via Verifier-Constrained Noised Space Entropy Maximization**

$$\pi^* \in \underset{\pi : p_0^\pi = p_0^{pre}}{\arg\max} \; \mathcal{L}(\mathbf{Q}^\pi) \coloneqq \int_0^1 \lambda_t \mathcal{G}_t(p_t^\pi) \, \mathrm{d}t \quad \text{subject to} \quad \underset{x \sim p_1^\pi}{\mathbb{E}} [v(x)] = 1 \tag{8}$$

Under this unifying formulation, $\mathcal{G}_t : \boldsymbol{P}(\mathcal{X}) \to \mathbb{R}$ is a functional over densities $p_t^\pi$ induced by flow $\pi$. We note that under general regularity assumptions, an optimal policy $\pi^*$ for Problem 8 is optimal also for the global and local flow expansion problems (see Eq. 5 and 7) if the functional $\mathcal{G}$ is defined as:

$$\underbrace{\mathcal{G}_t(p_t^\pi) = \mathcal{H}(p_t^\pi)}_{\text{Global Flow Expansion}} \qquad \underbrace{\mathcal{G}_t(p_t^\pi) = \mathcal{H}(p_t^\pi) - \alpha_t \, \mathrm{D}_{\mathrm{KL}}\big(p_t^\pi \,\|\, p_t^{\mathrm{pre}}\big)}_{\text{Local Flow Expansion } (\alpha_1 = \alpha)} \tag{9}$$

Before introducing FE, we first recall the standard notion of first variation of $\mathcal{G}$ over a space of probability measures (cf. Hsieh et al., 2019). A functional $\mathcal{G} \in \boldsymbol{F}(\mathcal{X})$, where $\mathcal{G} : \boldsymbol{P}(\mathcal{X}) \to \mathbb{R}$, has first variation at $\mu \in \boldsymbol{P}(\mathcal{X})$ if there exists a function $\delta\mathcal{G}(\mu) \in \boldsymbol{F}(\mathcal{X})$ such that for all $\mu' \in \boldsymbol{P}(\mathcal{X})$ it holds that:

$$\mathcal{G}(\mu + \epsilon\mu') = \mathcal{G}(\mu) + \epsilon\langle \mu', \delta\mathcal{G}(\mu)\rangle + o(\epsilon).$$

where the inner product is an expectation. Intuitively, $\delta\mathcal{G}(\mu)$ can be interpreted as an infinite-dimensional gradient over probability measures. Given this concept of first variation, FE solves Problem 8 by computing a process $\mathbf{Q}^k$ at each iteration $k \in [K]$, via the following mirror descent step:

**(MD Step) Constrained and Regularized Process Surprise Maximization**

$$\mathbf{Q}^k \in \underset{\mathbf{Q} : p_0 = p_0^{k-1}}{\arg\max} \langle \delta\mathcal{L}(\mathbf{Q}^{k-1}), \mathbf{Q}\rangle - \frac{1}{\gamma^k} D_{KL}\big(\mathbf{Q}\|\mathbf{Q}^{k-1}\big) \; \text{s.t.} \; \underset{x \sim p_1}{\mathbb{E}} [v(x)] = 1 \tag{10}$$

While the MD step in Eq. 10 might seem abstract, the following Lemma 4.1 hints at a more practical formulation of the above through the lens of stochastic optimal control (Fleming & Rishel, 2012).

**Lemma 4.1** (First Variation of Flow Process Functionals). *For objectives defined in the form of Eq. 8, we have:*

$$\langle \delta\mathcal{L}(\mathbf{Q}^k), \mathbf{Q}\rangle = \int_0^1 \lambda_t \; \underset{\mathbf{Q}}{\mathbb{E}} \big[\delta\mathcal{G}_t(p_t^k)\big] \, \mathrm{d}t. \tag{11}$$

Lemma 4.1 factorizes $\langle \delta\mathcal{L}(\mathbf{Q}^{k-1}), \mathbf{Q}\rangle$ into an integral over the flow process of terms $f_t(x) \coloneqq \lambda_t \delta\mathcal{G}_t(p_t^k)(x)$. Crucially, this time-decomposition allows to rewrite the MD step (Eq. 10) as the following standard constrained control-affine optimal control problem[2] (Domingo-Enrich et al., 2024):

**Constrained and Regularized Process Surprise Maximization via Fine-Tuning**

$$\min_\pi \; \mathbb{E}\left[\int_0^1 \frac{1}{2}\|\pi(X_t, t)\|^2 - f_t(X_t, t) \, \mathrm{d}t\right] s.t. \underset{x \sim p_1}{\mathbb{E}} [v(x)] = 1, \; \text{with } f_t(X_t, t) = \gamma_t \delta\mathcal{G}_t(p_t^k)(x)$$

Concretely, we compute a flow $\pi^k$ inducing $\mathbf{Q}^k$ (Eq. 10) via EXPANDTHENPROJECT (see Alg. 4), which decouples constrained optimization into sequential expansion and projection steps:

---

[2] We leave standard dynamical system constraints (e.g., Equation 13 Domingo-Enrich et al., 2024) as implicit.

---

**Algorithm 1** EXPANDTHENPROJECT

---

1: **Input:** $\pi^{k-1}$: current flow model, $\nabla_{x_t}\delta\mathcal{G}$: gradients of functional grad., $\gamma_k$: inverse update step-size, $\{\lambda_t\}_{t\in[0,1]}$: integral weighting coefficients , $v$: verifier, $\eta_k$: fine-tuning strength

2: **Expansion** step:

$$\tilde{\pi}^k \leftarrow \text{FINETUNINGSOLVER}(\pi^{k-1}, \nabla_{x_t}\delta\mathcal{G}_t, \lambda_t, \gamma_k) \qquad (12)$$

3: **Projection** step:

$$\pi^k \leftarrow \text{FINETUNINGSOLVER}(\tilde{\pi}^k, \log v, \eta_k) \qquad (13)$$

4: **Output:** Fine-tuned policy $\pi^k$

---

**Algorithm 2 F**low **E**xpander (FE)

---

1: **Input:** $\pi^{pre}$ : pre-trained flow model, $\{\alpha_t\}_{t\in[0,1]}$ : KL-regularization coefficients, $\{\gamma_k\}_{k=1}^K$ : inverse update step-sizes, $\{\lambda_t\}_{t\in[0,1]}$ : integral weighting coefficients, $v$: verifier, $\{\eta_k\}_{k=1}^K$: projection strengths

2: **Init:** $\pi_0 := \pi^{pre}$

3: **for** $k = 1, 2, \ldots, K$ **do**

4:     Set:

$$\nabla_{x_t}\delta\mathcal{G}_t(p_t^{k-1}) = \begin{cases} -s_t^{\pi^{k-1}} & \textbf{Global FE (G-FE)} \\ -s_t^{\pi} - \alpha_t\left(s_t^{\pi} - s_t^{\text{pre}}\right) & \textbf{Local FE (L-FE)} \end{cases} \qquad (14)$$

5:     Fine-tune $\pi_{k-1}$ into $\pi_k$ via Algorithm 4:

$$\pi_k \leftarrow \text{EXPANDTHENPROJECT}(\pi^{k-1}, \nabla_{x_t}\delta\mathcal{G}_t, \gamma_k, \{\lambda_t\}_{t\in[0,1]}, v, \eta_k)$$

6: **end for**

7: **Output:** policy $\pi := \pi_K$

---

**Expansion step.** The unconstrained expansion step is performed over the noised state space, which can be tackled by extending established control (or RL) based methods for fine-tuning with the running cost $f_t(X_t, t) = \gamma_t \delta\mathcal{G}_t(p_t^k)(x)$, effectively computing a process $\tilde{\mathbf{Q}}^k$ such that:

$$\tilde{\mathbf{Q}}^k \in \underset{\mathbf{Q}:p_0=p_0^{\text{pre}}}{\arg\max}\langle\delta\mathcal{L}(\mathbf{Q}^{k-1}), \mathbf{Q}\rangle - \frac{1}{\gamma_k}D_{\text{KL}}(\mathbf{Q}||\mathbf{Q}^{k-1}) \qquad (15)$$

**Projection step.** Given $\tilde{\mathbf{Q}}^k$, the projection step adapts the flow $\tilde{\pi}^k$ to enforce the constraint in Eq. 10 via reward-guided fine-tuning (e.g., Uehara et al., 2024a, Sec. 8.2):

$$\mathbf{Q}^k \in \underset{\mathbf{Q}:p_0=p_0^{\text{pre}}}{\arg\max}\underset{x\sim p_1}{\mathbb{E}}\left[\log v(x)\right] - D_{\text{KL}}(\mathbf{Q}||\tilde{\mathbf{Q}}^k) \qquad (16)$$

**Proposition 1.** *The* EXPANDTHENPROJECT *scheme in Alg. 4 solves optimization problem 10, i.e., it returns a flow model $\pi^k$ inducing a process $\mathbf{Q}^k$ that is a solution to 10. Formally, the following holds:*

$$\mathbf{Q}^k \in \underset{\mathbf{Q}:p_0=p_0^{k-1}}{\arg\max}\langle\delta\mathcal{L}(\mathbf{Q}^{k-1}), \mathbf{Q}\rangle - \frac{1}{\gamma^k}D_{KL}\left(\mathbf{Q}||\mathbf{Q}^{k-1}\right) \ s.t. \ \underset{x\sim p_1}{\mathbb{E}}\left[v(x)\right] = 1 \qquad (17)$$

Notice that this step could alternatively be performed via constrained reward-guided fine-tuning (e.g., Gutjahr et al., 2025). Finally, we present **F**low **E**xpander (FE) in Alg. 2, which effectively approximates the mirror descent scheme presented above by iteratively applying EXPANDTHENPROJECT.

**Complete algorithm execution.** FE takes as inputs: a pre-trained flow model $\pi^{pre}$, KL-regularization coefficients $\{\alpha_t\}_{t\in[0,1]}$, the number of iterations $K$, inverse step-sizes $\{\gamma_k\}_{k=1}^K$, integral weighting coefficients $\{\lambda_t\}_{t\in[0,1]}$, a verifier $v$, and a projection strength schedule $\{\eta_k\}_{k=1}^K$. At each $k \in [K]$, FE computes the gradient (w.r.t. $x$) of the first variation at the previous policy $\pi_{k-1}$, i.e., $\nabla_x \delta\mathcal{G}_t(p_1^{\pi^{k-1}})$ (line 4). Then, FE computes the flow model $\pi_k$ via the EXPANDTHENPROJECT scheme (see Alg. 4), which takes in input the current flow model $\pi^{k-1}$, the computed gradients, and the verifier $v$, and returns the updated flow model $\pi^k$. Ultimately, FE returns the final policy $\pi := \pi_K$.

**Closed-form gradient expressions.** FE operates using trajectory reward gradients $\nabla_{x_t}\delta\mathcal{G}_t(p_t^{\pi})$. In fact, while such rewards are difficult to estimate, their gradients admit close-form expressions (De Santi et al., 2025a) that can be approximated via available quantities:

$$\underbrace{\nabla_x\delta\mathcal{H}(p_t^{\pi}) = -s_t^{\pi}}_{\text{Global Flow Expansion}} \qquad \underbrace{\nabla_{x_t}\delta\mathcal{H}(p_t^{\pi}) - \alpha_t\,\nabla_x\delta\text{D}_{\text{KL}}\big(p_t^{\pi}\,\|\,p_t^{\text{pre}}\big) = -s_t^{\pi} - \alpha_t\left(s_t^{\pi} - s_t^{\text{pre}}\right)}_{\text{Local Flow Expansion }(\alpha_1 = \alpha)} \qquad (18)$$

---

**Algorithm 3** **N**oised **S**pace **E**xploration (NSE)

---

1: **Input:** $\pi^{pre}$ : pre-trained flow model, $\{\alpha_t\}_{t \in [0,1]}$ : KL-regularization coefficients, $\{\gamma_k\}_{k=1}^K$ : inverse update step-sizes, $\{\lambda_t\}_{t \in [0,1]}$ : integral weighting coefficients
2: **Init:** $\pi_0 := \pi^{pre}$
3: **for** $k = 1, 2, \ldots, K$ **do**
4:     Set:
$$\nabla_{x_t} \delta \mathcal{G}_t(p_t^{k-1}) = -s_t^{\pi} - \alpha_t \left( s_t^{\pi} - s_t^{\text{pre}} \right) \tag{19}$$

5:     **Expansion** step, fine-tune $\pi_{k-1}$ into $\pi_k$ via:
$$\pi^k \leftarrow \text{FINETUNINGSOLVER}(\pi^{k-1}, \nabla_{x_t} \delta \mathcal{G}_t, \lambda_t, \gamma_k) \tag{20}$$

6: **end for**
7: **Output:** policy $\pi := \pi_K$

---

and can simply be plugged into any first-order fine-tuning solver yielding a scalable method. Eq. 18 depends on the score $s_t^{\pi}(x) = \nabla \log p_t^{\pi}(x)$, obtained from the score network or flow field (Domingo-Enrich et al., 2024). Unlike prior flow-based exploration methods using only the divergent terminal score $s_1^{\pi}$ (De Santi et al., 2025a;b), FE uses scores across time and allows to downweight $t \to 1$ via $\lambda_t$.

Beyond verifier-constrained settings, noised-space exploration also yields a stronger unconstrained exploration scheme. We denote by **N**oised **S**pace **E**xploration (NSE) the variant of FE obtained by removing the projection step; its pseudocode is given in Alg. 3. As shown in Sec. 6, NSE stabilizes diffusion- and flow-based exploration in higher dimensions, substantially outperforming existing methods.

## 5  GUARANTEES FOR FLOW-EXPANDER

We aim to show that FE admits *provable guarantees* ensuring reliable behavior in practice. To this end, we leverage the flexible framework of *constrained mirror descent*, a classical optimization method that has recently found successful applications in sampling and generative modeling (Karimi et al., 2024; De Santi et al., 2025a;b). We analyze two regimes. First, an **idealized setting**, where each step of Eq. 10 can be computed *exactly* - leading to sharp step-size prescriptions and fast, polynomial convergence rates. Then, a **realistic setting**, where each MD step can only be solved *approximately* - for which we show asymptotic convergence to the optimal solution under mild noise and bias assumptions.

**Idealized setting.** We state that the exact updates case admits finite-time convergence guarantee:

> **Theorem 5.1** (Convergence guarantee in the idealized process-level setting). *Consider the objective $\mathcal{L}$ defined in Equation (8), and let $\lambda^\star := \int_0^1 \lambda_t \mathrm{d}t$. Let $\{\mathbf{Q}^k\}$ be the iterates generated by Equation (10) with $\gamma_k = 1/\lambda^\star$ for all $k \in [K]$. Then*
> $$\mathcal{L}(\mathbf{Q}^*) - \mathcal{L}(\mathbf{Q}^K) \leq \frac{\lambda^\star}{K} D_{KL}(\mathbf{Q}^* \,\|\, \mathbf{Q}^{pre}), \tag{21}$$
> *where $\mathbf{Q}^* \in \arg\max_{\mathbf{Q}} \mathcal{L}(\mathbf{Q})$.*

**General setting.** Recall that $\mathbf{Q}^k$ is the $k$-th iterate of FE. In realistic scenarios, however, Eq. 10 can only be solved approximately, so we interpret the update as *approximating* the idealized iteration:
$$\mathbf{Q}_{\sharp}^k \in \arg\max_{\mathbf{Q}:p_0 = p_0^{k-1}} \langle \delta \mathcal{L}(\mathbf{Q}^{k-1}), \mathbf{Q} \rangle - \tfrac{1}{\gamma^k} D_{KL}(\mathbf{Q} \,\|\, \mathbf{Q}^{k-1}) \quad \text{s.t.} \quad \mathbb{E}_{x \sim p_1}[v(x)] = 1. \tag{22}$$

To measure deviations from these idealized iterates, let $\mathcal{T}_k$ be the filtration up to step $k$, and decompose the oracle into its *bias* and *noise* components:
$$b_k := \mathbb{E}\left[ \delta \mathcal{L}(\mathbf{Q}^k) - \delta \mathcal{L}(\mathbf{Q}_{\sharp}^k) \,\middle|\, \mathcal{T}_k \right], \quad U_k := \delta \mathcal{L}(\mathbf{Q}^k) - \delta \mathcal{L}(\mathbf{Q}_{\sharp}^k) - b_k. \tag{23}$$

Here, $b_k$ captures systematic error while $U_k$ is conditionally mean-zero. Under mild assumptions on noise and bias (see Assumptions E.1 to E.2), we obtain the following guarantee.

> **Theorem 5.2** (Convergence guarantee in the general process-level setting (informal)). *Suppose the oracle satisfies finite-variance noise and vanishing bias, and let the step-sizes $\{\gamma_k\}$ follow the Robbins–Monro rule ($\sum_k \gamma_k = \infty$, $\sum_k \gamma_k^2 < \infty$). Then the iterates $\{\mathbf{Q}^k\}$ generated by FE satisfy*
> $$\mathbf{Q}^k \rightharpoonup \mathbf{Q}^* \quad a.s., \tag{24}$$
> *where $\mathbf{Q}^* \in \arg\max_{\mathbf{Q}} \mathcal{L}(\mathbf{Q})$.*

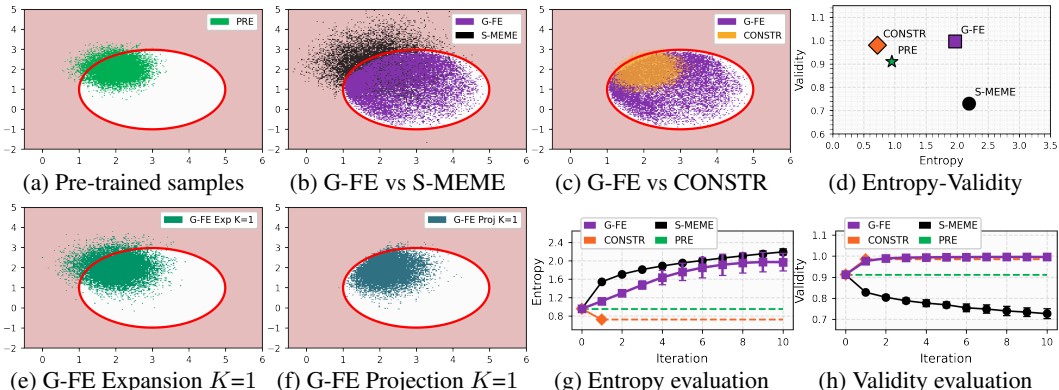

(a) Pre-trained samples    (b) G-FE vs S-MEME    (c) G-FE vs CONSTR    (d) Entropy-Validity

(e) G-FE Expansion $K$=1    (f) G-FE Projection $K$=1    (g) Entropy evaluation    (h) Validity evaluation

Figure 3: (top) **G**lobal **FE** (G-FE) expands the pre-trained flow model $\pi^{pre}$(3a) into $\pi^*$ (violet, 3b), increasing coverage (i.e., entropy), while preserving validity (i.e., red ellipse interior). Compared with the unconstrained exploration S-MEME method, and constrained generation (CONSTR), **G**lobal **FE** (G-FE) shows best-of-both-worlds behaviour: achieving near-optimal entropy and validity (Fig. 3d).

## 6 EXPERIMENTAL EVALUATION

We analyze the ability of **G**lobal **FE** (G-FE) and **L**ocal **FE** (L-FE) to expand flow model densities while preserving validity of generated samples, and compare their performance against recent flow-based exploration methods, namely FDC (De Santi et al., 2025b), and S-MEME (De Santi et al., 2025a), as well as a standard constrained generation scheme, denoted by CONSTR (Sec. 8.2 Uehara et al., 2024a). We present experiments on two visually interpretable settings, followed by a molecular design task aiming to increase conformer diversity (more details are provided in Apx. G).

**Global Flow Expansion via Strong Verifier.** We run G-FE on a pre-trained model $\pi^{pre}$ to globally expand its density $p_1^{pre}$ over the valid design space (red ellipse in Figs. 3a-3c, 3e-3f). As shown in Fig. 3b and 3c, G-FE (violet) run with $\eta = 2$ and $K = 10$, expands into previously uncovered areas (lower right), staying within the valid region. In comparison, S-MEME (black, Fig. 3b) predictably fails to restrict density to the valid region (light red area). Symmetrically, CONSTR (see Fig. 3c, orange) confines density to the valid space but fails to expand it. Fig. 3d shows that G-FE explores nearly as much as S-MEME (i.e., 1.97 vs. 2.17 entropy), while retaining significantly higher validity: 0.99 against 0.73 of S-MEME. Remarkably, G-FE preserves the same degree of validity of CONSTR while exploring significantly more (1.97 versus 0.72 entropy). Figs. 3e-3f show the first expand-then-project steps of G-FE and Figs. 3g-3h show entropy and validity estimates with 95% CI over 5 seeds for G-FE and all baselines. In summary, G-FE achieves both near-optimal exploration and validity.

**Local Flow Expansion via Weak Verifier.** We consider a pre-trained flow model $\pi^{pre}$ whose density $p_1^{pre}$ is concentrated in a central high-density region, with low-probability *promising* modes on either side (see Fig. 5a). Crucially, while the two right-most modes are valid, the left one is not. We fine-tune $\pi^{pre}$ via L-FE for $K = 8$ iterations and $\alpha = 0.99$ to expand its induced density over diverse modes - i.e., perform *mode discovery* (De Santi et al., 2025b; Morshed & Boddeti, 2025). As shown in Fig. 5b FDC, a KL-regularized entropy maximization scheme, predictably increases diversity over plausible modes by redistributing density to the invalid left one. L-FE, however, leverages a weak verifier (gray circled area in Fig. 5c) to prevent allocating more density to that invalid region, and even removes density from that region. 5a, top). Effectively, L-FE uses the weak verifier to perform a form of mode selection, i.e., filtering out invalid modes during the expansion process. As shown in Fig. 4, **L**ocal **FE** (L-FE) achieves high entropy (i.e., 1.67 versus 1.17 and 1.58 of L-FE), while preserving high validity, namely 0.89 compared to 0.74 of FDC, almost fully preserving the prior model's validity of 0.9.

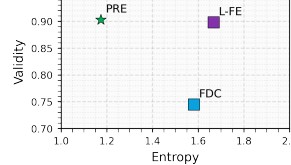

Figure 4: Entropy-Validity

**L-FE increases molecular conformer diversity for de-novo design on QM9.** In this experiment, we aim to increase diversity of molecular conformers in a molecular design task. We run FE on FlowMol CTMC (Dunn & Koes, 2024) pre-trained on QM9 dataset (Ramakrishnan et al., 2014). Our weak verifier is a filter excluding molecules for which any two atoms are closer than 0.975 Ångstroms (Å), and validity is evaluated via RDKit (RDKit) sanitization paired with the aforementioned check.

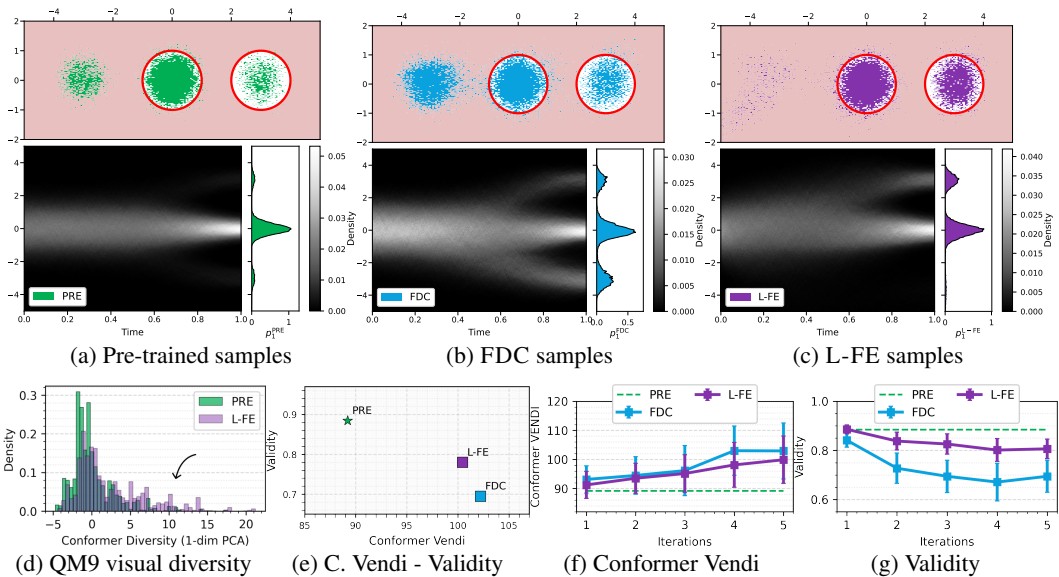

(a) Pre-trained samples      (b) FDC samples      (c) L-FE samples

(d) QM9 visual diversity    (e) C. Vendi - Validity    (f) Conformer Vendi    (g) Validity

Figure 5: (top) L-FE (yellow, 5c) expands the pre-trained flow model $\pi^{pre}$ (green, 5a) over promising yet verifier-filtered modes, while FDC (blue, 5b) expands $\pi^{pre}$ over all plausible modes leading to increased density in invalid regions (left mode in Fig. 5b). (bottom) FE increases visual (5d), and quantitative diversity (5f), while preserving higher validity than FDC (5e-5g)

We evaluate diversity of molecular conformers by a *conformer* VENDI (Friedman & Dieng, 2022) metric (see Apx. G.2) capturing diversity over sampled conformers via their fingerprints. L-FE, run for $K = 5$ iterations and $\alpha = 9$, quantitatively increases diversity compared to the pretrained model (Fig 5e, VENDI of 100 vs 89). This is visually shown in Fig 5d, a histogram plot of a 1-dim PCA projection of molecular fingerprints (see Apx. G for further details). In particular, L-FE (violet) expands the pre-trained flow model to explore promising and verifier-certified modes of the pre-trained model density (see Fig. 5d). Crucially, L-FE achieves a similar degree of conformer diversity (100 vs 103) to FDC, an unconstrained exploration scheme, while preserving significantly higher sample validity, i.e., 81% vs 69%, as shown in Figs. 5e, 5f, and 5g.

**L-FE increases molecular conformer diversity for de-novo design on GEOM-Drugs.** In this experiment, we aim to increase the diversity of generated molecular conformers in a molecular design task with drug-like molecules. We run FE on FlowMol CTMC (Dunn & Koes, 2024) pre-trained on GEOM-Drugs (Axelrod & Gomez-Bombarelli, 2022). As in the previous experimental setting, the weak verifier employed is a filter excluding molecules for which any two atoms are closer than 0.975 Ångstroms (Å), and validity is evaluated via the RDKit (RDKit) sanitization operation paired with the aforementioned check. We evaluate diversity of molecular conformers by a *conformer* VENDI (Friedman & Dieng, 2022) metric (see Apx. G.2) capturing diversity over sampled conformers via their fingerprints. L-FE, run for $K = 3$ iterations, $\alpha = 1/9$, and $\eta = 5$, achieves higher diversity (529 vs 476) and validity (82% vs 72%) than the pre-trained model, as shown in Fig. 6a. Similarly, L-FE induces higher diversity (529 vs 508) than FDC, a recent diffusion-based unconstrained exploration method (De Santi et al., 2025b), while preserving significantly higher sample validity, i.e., 82% vs 66%, as shown in Fig. 6a.

Moreover, within Apx. G.5.4, we report an ablation study for the proposed method parameters. Note that L-FE performs consistently better than NSE. Since NSE corresponds to L-FE with $\eta = 0$, this result illustrates the working mechanism and importance of the projection step (i.e., $\eta > 0$). Interestingly, G-FE, which is equal to L-FE with $\alpha_t = 0$, shows performance on par with L-FE for very conservative parameters ($\gamma = 0.0002$, $K = 3$), while gradually degrading for less conservative parametrizations. This behaviour is likely due to the implicit KL-regularization between iterates within the mirror descent update step (see Eq. 10), which implies prior regularization for small $K$.

**NSE achieves higher exploration performance against current methods.** We evaluate NSE, the unconstrained exploration variant of L-FE obtained by removing the projection step (see Alg. 3 in Sec. 4 for further details), to perform flow-based design space (unconstrained) exploration. We consider FlowMol CTMC (Dunn & Koes, 2024) pre-trained on GEOM-Drugs dataset (Axelrod & Gomez-Bombarelli, 2022), and report in Fig. 6b the results for NSE with $K = 3$ and $\alpha = 0$. We observe that

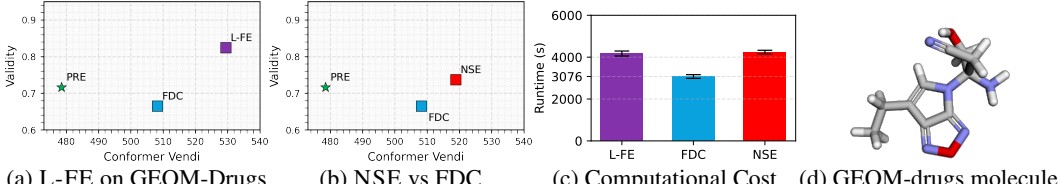

(a) L-FE on GEOM-Drugs   (b) NSE vs FDC   (c) Computational Cost   (d) GEOM-drugs molecule

Figure 6: (6a) L-FE (violet) achieves higher diversity and validity than FDC on GEOM-Drugs. (6b) NSE, the unconstrained variant of L-FE, exhibits superior performance compared with FDC, a state-of-the-art diffusion-based unconstrained exploration method. (6c) Computational cost comparison of L-FE, FDC, and NSE. (6d) Representative drug-like molecule generated via L-FE.

NSE achieves consistently higher diversity (i.e., 519 vs 508) and validity (i.e., 74% vs 66%) against FDC (De Santi et al., 2025b), a state-of-the-art method for flow-based unconstrained exploration.

**L-FE and NSE have computational costs comparable to current exploration schemes.** We report in Fig. 6c a comparison of computational cost, measured via the method runtime (seconds [s]) of L-FE and NSE compared against FDC (De Santi et al., 2025b). One can notice that although the schemes proposed within this work (i.e., L-FE and NSE) perform exploration over the entire flow process noised state space, they do not incur in significantly higher computational cost compared with FDC.

## 7 RELATED WORK

**Diffusion and flow based design space exploration**   Recent works introduced methods for flow based design space exploration via maximization of entropy functionals (De Santi et al., 2025a;b) or approximations (Celik et al., 2025). While these methods explore by leveraging information from a prior model, FE directs exploration either $(i)$ exclusively via a verifier (i.e., global expansion, see 5), or $(ii)$ combining verifier information with prior validity cues (i.e., local expansion, see 7). Moreover, while current schemes explore only the last time-step state space, we lift the exploration task to the entire flow process, providing a principled solution to the score divergence problem mentioned in Sec. 4.

**Maximum state entropy exploration.** Maximum state entropy exploration, introduced by Hazan et al. (2019), tackles the pure-exploration problem of maximizing the entropy of the state distribution induced by a policy over a dynamical system's state space (e.g., Lee et al., 2019; Mutti et al., 2021; Guo et al., 2021; De Santi et al., 2024a). The flow expansion problems (Eq. 5 and 7) are closely related, with $p_1^\pi$ representing the state distribution induced by policy $\pi$ over a subset of the flow process state space (i.e., for time-step $t = 1$). Recent studies have tackled maximum entropy exploration with finite sample budgets (e.g., De Santi et al., 2024b; Prajapat et al., 2023; Mutti et al., 2023; 2022b;a), which could be relevant for future work, e.g., design space exploration under a limited samples constraint.

**Sample diversity in diffusion models generation.**   A well-known limitation of flow-based generation is limited sample diversity. This problem has been recently addressed by numerous studies (e.g., Corso et al., 2023; Um et al., 2023; Kirchhof et al., 2024; Sadat et al., 2024; Um & Ye, 2024; Klarner et al., 2024). Crucially, such methods are complementary to ours. In fact, they can be applied to promote diverse sampling from the expanded model produced by FE. In particular, whereas these works aim to maximize diversity of a fixed diffusion model or flow model, we aim to sequentially fine-tune a pre-trained flow model so that its induced density is permanently expanded over the valid design space. Moreover, our formulations (Eq. 5, 7) and FE scheme increase diversity while integrating validity signal from a chosen verifier.

## 8 CONCLUSION

This work tackles the fundamental challenge of leveraging a verifier (e.g., an atomic bonds checker), to expand a pre-trained model's density beyond regions of high data availability, while preserving validity of the generated samples. To this end, we introduce notions of *strong* and *weak* verifiers and cast *global* and *local flow expansion* as probability-space optimization problems. We present **F**low **E**xpander (FE), a scalable mirror-descent scheme that *provably* solves both problems via verifier-constrained entropy maximization over the flow process noised state space. We provide a thorough analysis showing convergence guarantees for FE under idealized and general assumptions by employing recent mirror-flow theory. Ultimately, we empirically evaluate our method on both illustrative settings, and a molecular design task showcasing the ability of FE to increase molecular conformer diversity while preserving better levels of validity than current flow and diffusion-based exploration methods.

## 9 ACKNOWLEDGEMENTS

This publication was supported by the ETH AI Center doctoral fellowship to Riccardo De Santi. The project has received funding from the Swiss National Science Foundation under NCCR Catalysis grant number 180544 and NCCR Automation grant agreement 51NF40 180545.

## 10 REPRODUCIBILITY STATEMENT

We acknowledge that our work is documented sufficiently to ensure reproducibility of our results. We provide implementation details of all algorithms and procedures, such as: complete pseudocode in Appendix F, as well as hyperparameter choices and hardware requirements in Appendix Section G. We also give a detailed account of our experimental setup in Section 6, including explanations of metrics and procedures used to evaluate our algorithms in Appendix G. Finally, our implemented version of FE, leverages the well-established Adjoint Matching (Domingo-Enrich et al., 2024) method as an oracle.

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

# A APPENDIX

## CONTENTS

## B  DERIVATION OF GRADIENTS OF FIRST VARIATION

In this section we present derivations of the results in equation 14 relating the gradient of the first variation of the trajectory rewards to the score function. We derive the result for L-FE, the result for G-FE follows as a subcase.

First, recall the trajectory rewards for L-FE:

$$
\nabla_{x_t} \delta \mathcal{G}_t(p_t^\pi) = \nabla_{x_t} \delta \big( \mathcal{H}(p_t^\pi) - \alpha_t \mathcal{D}_{\mathrm{KL}}(p_t^\pi || p_t^{\mathrm{pre}}) \big) \tag{25}
$$

$$
= \nabla_{x_t} \delta \mathcal{H}(p_t^\pi) - \alpha_t \nabla_{x_t} \delta \mathcal{D}_{\mathrm{KL}}(p_t^\pi || p_t^{\mathrm{pre}}) . \qquad \text{(by linearity)} \tag{26}
$$

Thus it suffices to show derivations for $\nabla_{x_t} \delta \mathcal{H}(p_t^\pi)$ and $\nabla_{x_t} \delta \mathcal{D}_{\mathrm{KL}}(p_t^\pi || p_t^{\mathrm{pre}})$. Starting with the entropy functional, recalling its definition as $\mathcal{H}(p_t^\pi) = - \int_0^1 p_t^\pi(x) \log p_t^\pi(x) dx$ we have:

$$
\nabla_{x_t} \delta \mathcal{H}(p_t^\pi) = \nabla_{x_t}(1 - \log p_t^\pi) \tag{27}
$$

$$
= -\nabla_{x_t} \log p_t^\pi \tag{28}
$$

$$
= -s_t^\pi \tag{29}
$$

Similarly for the second term:

$$
\nabla_{x_t} \delta \mathcal{D}_{\mathrm{KL}}(p_t^\pi || p_t^{\mathrm{pre}}) = \nabla_{x_t} \int p_t^\pi \log p_t^\pi - \tag{30}
$$

$$
= \nabla_{x_t}(\log p_t^\pi - 1 - \log p_t^{\mathrm{pre}}) \tag{31}
$$

$$
= s_t^\pi - s_t^{\mathrm{pre}} \tag{32}
$$

## C   PROOF OF PROPOSITION 1

In this section we show that the optimization problem in 10 can be decomposed into an unconstrained expansion step followed by a projection into the constrained set. We start by defining the following processes:

$$\mathbf{Q}^k \in \underset{\mathbf{Q}:p_0=p_0^{k-1}}{\arg\max} \langle \delta\mathcal{L}(\mathbf{Q}^{k-1}), \mathbf{Q}\rangle - \frac{1}{\gamma_k}\mathcal{D}_{\mathrm{KL}}(\mathbf{Q}||\mathbf{Q}^{k-1}) \quad \text{s.t.} \quad \underset{x\sim q_1}{\mathbb{E}}[v(x)] \tag{33}$$

$$\tilde{\mathbf{Q}}^k \in \underset{\mathbf{Q}:p_0=p_0^{k-1}}{\arg\max} \langle \delta\mathcal{L}(\mathbf{Q}^{k-1}), \mathbf{Q}\rangle - \frac{1}{\gamma_k}\mathcal{D}_{\mathrm{KL}}(\mathbf{Q}||\mathbf{Q}^{k-1}) \tag{34}$$

$$\bar{\mathbf{Q}}^k \in \underset{\mathbf{Q}:p_0=p_0^{k-1}}{\arg\min} \mathcal{D}_{\mathrm{KL}}(\mathbf{Q}||\tilde{\mathbf{Q}}^k) \quad \text{s.t.} \quad \underset{x\sim p_1}{\mathbb{E}}[v(x)] = 1 \tag{35}$$

$$\hat{\mathbf{Q}}^k \in \underset{\mathbf{Q}:p_0=p_0^{k-1}}{\arg\max} \underset{x\sim p_1}{\mathbb{E}}[\log v(x)] - \frac{1}{\gamma_k}\mathcal{D}_{\mathrm{KL}}(\mathbf{Q}||\mathbf{Q}^{k-1}) \tag{36}$$

letting $p_t^k, \tilde{p}_t^k, \bar{p}_t^k, \hat{p}_t^k$ refer to their respective marginal densities a time $t$. Note that $\tilde{\mathbf{Q}}^k$ is the output of the projection step in 15, and that $\hat{\mathbf{Q}}^k$ is the output of the projection step in 16. The following Lemma asserts that solving the optimization problem in equation 10 is equivalent to solving the expansion step of 34 followed by the formal information projection step of 35.

**Lemma C.1.** *Let $\mathbf{Q}^{k-1}$ be the process associated with the previous iterate $\pi^{k-1}$, and let $\tilde{\mathbf{Q}}^k$ and $\bar{\mathbf{Q}}^k$ be defined as above. Then $\bar{\mathbf{Q}}^k = \mathbf{Q}^k$.*

*Proof.* First, note that the processes $\mathbf{Q}^{k-1}$ and $\tilde{\mathbf{Q}}_t^k$ satisfy the following relationship (see e.g. Domingo-Enrich et al. (2024) equation 22):

$$\log \frac{d\tilde{\mathbf{Q}}^k}{d\mathbf{Q}^{k-1}}(X) = \gamma_k \delta\mathcal{L}(\mathbf{Q}^{k-1})(X) + \text{const}. \tag{37}$$

which implies the following equality for an arbitrary process $q$ (taking the expectation and rearranging):

$$\langle \delta\mathcal{L}(\mathbf{Q}^{k-1}), \mathbf{Q}\rangle - \gamma_k D_{\mathrm{KL}}(\mathbf{Q}||\mathbf{Q}^{k-1}) = \gamma_k D_{\mathrm{KL}}(\mathbf{Q}||\tilde{\mathbf{Q}}^k) - \text{const}. \tag{38}$$

Therefore the equation below holds for any arbitrary set of processes $A$:

$$\underset{\mathbf{Q}\in A}{\arg\max} \langle \delta\mathcal{L}(\mathbf{Q}^{k-1}), q\rangle - \frac{1}{\gamma_k}D_{\mathrm{KL}}(\mathbf{Q}||\mathbf{Q}^{k-1}) = \underset{q\in A}{\arg\min} D_{\mathrm{KL}}(\mathbf{Q}||\tilde{\mathbf{Q}}^k) \tag{39}$$

and thus also holds for the set $A = \{\mathbf{Q} \text{ s.t. } p_0 = p_0^{k-1} \text{ and } \mathbb{E}_{x\sim p_1}[v(x)] = 1\}$: the set of feasible solutions to 10.

$\square$

Finally, the following Lemma reformulates the information projection step in 35 as the fine-tuning objective in 34:

**Lemma C.2.** *Let $\hat{\mathbf{Q}}^k$ and $\bar{\mathbf{Q}}^k$ be defined as above. Then $\hat{\mathbf{Q}}^k = \bar{\mathbf{Q}}^k$*

*Proof.* Recall the definition of $\hat{\mathbf{Q}}^k$:

$$\hat{\mathbf{Q}}^k \in \underset{\mathbf{Q}:p_0=p_0^{k-1}}{\arg\max} \underset{x\sim p_1}{\mathbb{E}}[\log v(x)] - \frac{1}{\gamma_k}\mathcal{D}_{\mathrm{KL}}(\mathbf{Q}||\mathbf{Q}^{k-1}) \tag{40}$$

and note that the expectation in the first term is finite only if $v(x) \neq 0$, $p_1 -$ a.s., in which case it vanishes. Thus the maximizer must belong to the set $\left\{ \mathbf{Q} : p_0 = p_0^{k-1}, \; \mathbb{E}_{x \sim p_1}[v(x)] = 1 \right\}$ effectively turning the first term into a constraint. $\qquad \square$

## D   PROOF FOR THEOREM 5.1

**Theorem 5.1** (Convergence guarantee in the idealized process-level setting). *Consider the objective $\mathcal{L}$ defined in Equation (8), and let $\lambda^\star := \int_0^1 \lambda_t \mathrm{d}t$. Let $\{\mathbf{Q}^k\}$ be the iterates generated by Equation (10) with $\gamma_k = 1/\lambda^\star$ for all $k \in [K]$. Then*

$$\mathcal{L}(\mathbf{Q}^*) - \mathcal{L}(\mathbf{Q}^K) \leq \frac{\lambda^\star}{K} D_{KL}(\mathbf{Q}^* \,\|\, \mathbf{Q}^{pre}), \tag{21}$$

*where $\mathbf{Q}^* \in \arg\max_{\mathbf{Q}} \mathcal{L}(\mathbf{Q})$.*

*Proof.* Fix an initial reference measure $\bar{\mathbf{Q}} := \mathbf{Q}^0$, and define the function

$$\mathcal{Q}(\mathbf{Q}) := D_{\mathrm{KL}}(\mathbf{Q} \,\|\, \bar{\mathbf{Q}}), \tag{41}$$

which measures the Kullback–Leibler divergence of $\mathbf{Q}$ from this reference. This choice of $\mathcal{Q}$ will serve as the *reference function* in the framework of *mirror descent with relative smoothness* (Bauschke et al., 2017; Lu et al., 2018). The key point is that the objective $\mathcal{L}$ in Equation (8) is not necessarily smooth in the classical sense, but it is $\lambda^\star$-*smooth relative to* $\mathcal{Q}$.

To formalize this, let $D_{\mathcal{Q}}(\mathbf{Q}, \mathbf{Q}')$ denote the *Bregman divergence* generated by $\mathcal{Q}$. By definition,

$$D_{\mathcal{Q}}(\mathbf{Q}, \mathbf{Q}') = \mathcal{Q}(\mathbf{Q}) - \mathcal{Q}(\mathbf{Q}') - \langle \delta\mathcal{Q}(\mathbf{Q}'), \mathbf{Q} - \mathbf{Q}' \rangle.$$

A direct computation shows that when $\mathcal{Q}$ is the KL divergence from a fixed reference measure, the Bregman divergence reduces exactly to another KL divergence:

$$D_{\mathcal{Q}}(\mathbf{Q}, \mathbf{Q}') = D_{\mathrm{KL}}(\mathbf{Q} \,\|\, \mathbf{Q}').$$

This equivalence will allow us to leverage classical properties of relative entropy in the convergence analysis.

Next, consider the mirror descent iterates $\{\mathbf{Q}^k\}$ for minimizing $(-\mathcal{L})$[3]. By the definition of relative smoothness, we have

$$(-\mathcal{L})(\mathbf{Q}^k) \leq (-\mathcal{L})(\mathbf{Q}^{k-1}) + \langle \delta(-\mathcal{L})(\mathbf{Q}^{k-1}), \mathbf{Q}^k - \mathbf{Q}^{k-1} \rangle + \lambda^\star D_{\mathcal{Q}}(\mathbf{Q}^k, \mathbf{Q}^{k-1}). \tag{42}$$

Here, the first inequality follows directly from the $\lambda^\star$-*smoothness of* $(-\mathcal{L})$ *relative to* $\mathcal{Q}$, as defined in Equation (41). Intuitively, this is a generalization of the standard quadratic upper bound used in classical smooth optimization, but with the Bregman divergence replacing the squared Euclidean norm.

We can refine this bound further by applying the *three-point inequality* of the Bregman divergence (Lu et al., 2018, Lemma 3.1). Let us define a linearized function

$$\phi(\mathbf{Q}) := \frac{1}{\lambda^\star} \langle \delta(-\mathcal{L})(\mathbf{Q}^{k-1}), \mathbf{Q} - \mathbf{Q}^{k-1} \rangle,$$

and let $z = \mathbf{Q}^{k-1}$, $z^+ = \mathbf{Q}^k$. Then the three-point identity gives

$$\langle \delta(-\mathcal{L})(\mathbf{Q}^{k-1}), \mathbf{Q}^k - \mathbf{Q}^{k-1} \rangle \leq \langle \delta(-\mathcal{L})(\mathbf{Q}^{k-1}), \mu - \mathbf{Q}^{k-1} \rangle + \lambda^\star D_{\mathcal{Q}}(\mu, \mathbf{Q}^{k-1}) - \lambda^\star D_{\mathcal{Q}}(\mu, \mathbf{Q}^k), \tag{43}$$

for any reference point $\mu$. Combining Equation (42) and Equation (43) yields

$$(-\mathcal{L})(\mathbf{Q}^k) \leq (-\mathcal{L})(\mathbf{Q}^{k-1}) + \langle \delta(-\mathcal{L})(\mathbf{Q}^{k-1}), \mu - \mathbf{Q}^{k-1} \rangle + \lambda^\star D_{\mathcal{Q}}(\mu, \mathbf{Q}^{k-1}) - \lambda^\star D_{\mathcal{Q}}(\mu, \mathbf{Q}^k). \tag{44}$$

Finally, we can telescope this inequality over $k = 1, \ldots, K$. Using the monotonicity of $(-\mathcal{L})(\mathbf{Q}^k)$ along the iterates and the non-negativity of the Bregman divergence $D_{\mathcal{Q}}$, we obtain (Lu et al., 2018):

$$\sum_{k=1}^{K} \left( (-\mathcal{L})(\mathbf{Q}^k) - (-\mathcal{L})(\mu) \right) \leq \lambda^\star D_{\mathcal{Q}}(\mu, \mathbf{Q}^0) - \lambda^\star D_{\mathcal{Q}}(\mu, \mathbf{Q}^K) \leq \lambda^\star D_{\mathcal{Q}}(\mu, \mathbf{Q}^0), \tag{45}$$

for any $\mathbf{Q}$. Dividing both sides by $K$ and rearranging gives a simple *ergodic convergence rate*:

$$(-\mathcal{L})(\mathbf{Q}^K) - (-\mathcal{L})(\mathbf{Q}) \leq \frac{\lambda^\star D_{\mathcal{Q}}(\mathbf{Q}, \mathbf{Q}^0)}{K}, \tag{46}$$

which shows that the iterates converge at an $O(1/K)$ rate in terms of the relative entropy. $\qquad\square$

---

[3]We adopt the standard convention of convex *minimization* rather than concave maximization, which explains the negative sign in the formulation.

# E    PROOF FOR THEOREM 5.2

To establish our main convergence theorem, we impose a few auxiliary assumptions that are widely used in the analysis of stochastic approximation and gradient flows. These assumptions are mild and typically satisfied in practical applications.

**Assumption E.1** (Precompactness of Dual Iterates). *The sequence of dual variables $\{\delta\mathcal{Q}(\mathbf{Q}^k)\}_k$ is precompact in the $L_\infty$ topology.*

Precompactness ensures that the interpolated trajectories of the dual iterates remain within a bounded region in function space. This property is crucial for applying convergence results based on asymptotic pseudotrajectories, and similar precompactness assumptions have appeared in the literature on stochastic approximation and continuous-time interpolations of discrete dynamics (Benaïm, 2006; Hsieh et al., 2019; Mertikopoulos et al., 2024).

In our finite-dimensional parameter space, (E.1) essentially requires that the sequence of iterates produced by the solver remains in a bounded set. This is a very mild requirement: it is satisfied as soon as the solver does not diverge numerically (e.g., no exploding parameters or NaNs), which is exactly what we observe in all our experiments. Moreover, standard practices such as bounded initialization, weight decay, and gradient clipping can be viewed as explicit mechanisms that enforce this boundedness.

**Assumption E.2** (Noise and Bias Control). *The stochastic approximations in the updates satisfy, almost surely, the following conditions:*

$$\|b_k\|_\infty \to 0, \tag{47}$$

$$\sum_k \mathbb{E}\big[\gamma_k^2\big(\|b_k\|_\infty^2 + \|U_k\|_\infty^2\big)\big] < \infty, \tag{48}$$

$$\sum_k \gamma_k\|b_k\|_\infty < \infty. \tag{49}$$

These conditions are standard in the Robbins–Monro framework (Robbins & Monro, 1951; Benaïm, 2006; Hsieh et al., 2019). They guarantee that the bias of the stochastic updates vanishes asymptotically, and that the cumulative effect of the noise remains controlled. Together, they ensure that the stochastic perturbations do not prevent convergence of the iterates to the optima of the target objective.

With these assumptions in place, we are ready to restate the main result and present its proof.

**Theorem E.1** (Convergence guarantee in the general trajectory setting (rigorous)). *Suppose the oracle satisfies Assumptions E.1 to E.2, and let the step-sizes $\{\gamma_k\}$ follow the Robbins–Monro rule ($\sum_k \gamma_k = \infty$, $\sum_k \gamma_k^2 < \infty$). Then the iterates $\{\mathbf{Q}^k\}$ generated by FE satisfy*

$$\mathbf{Q}^k \rightharpoonup \mathbf{Q}^*  \quad a.s., \tag{50}$$

*where $\mathbf{Q}^* \in \arg\max_{\mathbf{Q}} \mathcal{L}(\mathbf{Q})$.*

*Proof.* As in the proof of Theorem 5.1, fix an initial reference measure

$$\bar{\mathbf{Q}} := \mathbf{Q}^0,$$

and define the relative entropy functional

$$\mathcal{Q}(\mathbf{Q}) := D_{\mathrm{KL}}\big(\mathbf{Q} \,\big\|\, \bar{\mathbf{Q}}\big). \tag{51}$$

Correspondingly, we introduce the initial dual variable

$$\mathbf{h}_0 := \delta\mathcal{Q}(\mathbf{Q}^0) = -\log \frac{\mathrm{d}\mathbf{Q}^0}{\mathrm{d}\bar{\mathbf{Q}}},$$

where $\frac{\mathrm{d}\mathbf{Q}}{\mathrm{d}\bar{\mathbf{Q}}}$ denotes the Radon–Nikodym derivative of $\mathbf{Q}$ with respect to $\bar{\mathbf{Q}}$. This dual representation encodes the convex geometry of the problem.

**Continuous-time mirror flow.** We now consider the continuous-time mirror flow dynamics

$$\begin{cases} \dot{\mathbf{h}}_t = \delta(-\mathcal{L})(\mathbf{Q}_t), \\ \mathbf{Q}_t = \delta\mathcal{Q}^\star(\mathbf{h}_t), \end{cases} \tag{MF}$$

where $\mathcal{Q}^\star$ denotes the Fenchel conjugate of the relative entropy functional. Explicitly, we recall that

$$\mathcal{Q}^\star(\mathbf{h}) = \log_{\bar{\mathbf{Q}}} \mathbb{E}\big[e^{\mathbf{h}}\big],$$

which follows from the variational characterization of the Kullback–Leibler divergence (Hsieh et al., 2019; Hiriart-Urruty & Lemaréchal, 2004).

**Discrete-to-continuous interpolation.** To connect the discrete algorithm with the flow equation MF, we introduce an interpolation of the iterates. Define the linearly interpolated process $\mathbf{h}(t)$ by

$$\mathbf{h}(t) = \mathbf{h}^k + \frac{t - \tau^k}{\tau^{k+1} - \tau^k}\big(\mathbf{h}^{k+1} - \mathbf{h}^k\big), \quad \mathbf{h}^k = \delta\mathcal{Q}(\mathbf{Q}^k), \quad \tau^k = \sum_{r=0}^{k} \alpha_r, \tag{Int}$$

where $\alpha_r$ are the step sizes. This construction yields a continuous-time trajectory $\{\mathbf{h}(t)\}_{t \geq 0}$ that faithfully tracks the discrete iterates in the limit of vanishing step sizes.

**Asymptotic pseudotrajectories.** We recall the notion of an asymptotic pseudotrajectory (APT), which provides the precise mathematical bridge between discrete stochastic processes and deterministic flows.

Let $\Theta$ denote the flow map associated with equation MF; that is, $\Theta_h(\mathbf{f})$ is the solution of equation MF at time $h$ when initialized at $\mathbf{f}$.

**Definition 3** (Asymptotic Pseudotrajectory (APT)). *A trajectory $\mathbf{h}(t)$ is called an asymptotic pseudotrajectory (APT) of equation MF if, for every finite horizon $T > 0$,*

$$\lim_{t \to \infty} \sup_{0 \leq h \leq T} \|\mathbf{h}(t+h) - \Theta_h(\mathbf{h}(t))\|_\infty = 0.$$

Intuitively, this condition requires that the interpolated sequence asymptotically shadows the exact flow on every bounded time interval.

**Limit set characterization.** The central result of Benaïm (2006) asserts that the long-term behavior of an APT is governed by the internally chain transitive (ICT) sets of the limiting flow.

**Theorem E.2** (APT Limit Set Theorem (Benaïm, 2006, Thm. 4.2)). *If $\mathbf{h}(t)$ is a precompact APT of equation MF, then almost surely its limit set lies within the set of internally chain-transitive (ICT) points of the flow.*

**Reduction of the convergence proof.** With these tools, the convergence analysis reduces to verifying two key claims:

(C1) Under Assumptions E.1 to E.2, the interpolated sequence $\{\mathbf{h}(t)\}$ indeed forms a precompact APT of equation MF.

(C2) The set of ICT points of the flow equation MF coincides with the set of stationary points of $\mathcal{L}$.

**Verification of Claim (C1).** Precompactness follows directly from Assumption E.1, which guarantees uniform tightness of the sequence of measures and hence compactness of their trajectories in the weak topology. In addition, standard arguments from stochastic approximation (Hsieh et al., 2019; Benaïm, 2006; Mertikopoulos et al., 2024) yield the following quantitative estimate: for every finite horizon $T > 0$, there exists a constant $C(T) > 0$ such that

$$\sup_{0 \leq h \leq T} \|\mathbf{h}(t+h) - \Theta_h(\mathbf{h}(t))\| \leq C(T)\big[\Delta(t-1, T+1) + b(T) + \gamma(T)\big],$$

where $\Delta(t-1, T+1)$ denotes the cumulative effect of noise over the interval $[t-1, t+T+1]$, while $b(T)$ and $\gamma(T)$ capture, respectively, the bias and step-size contributions. This bound quantifies the deviation of the interpolated process from the deterministic mirror flow equation MF.

**APT approximation.** Under the noise and bias conditions of Assumption E.2, both perturbations vanish asymptotically:

$$\lim_{t \to \infty} \Delta(t-1, T+1) = \lim_{t \to \infty} b(T) = 0,$$

uniformly over bounded horizons $T$. Consequently, the discrepancy in the above bound vanishes in the limit, and the interpolated process $\mathbf{h}(t)$ shadows the continuous-time flow arbitrarily well.

Altogether, these arguments show that $\mathbf{h}(t)$ is indeed a precompact asymptotic pseudotrajectory of the mirror flow.

**Verification of Claim (C2).** The flow equation MF is precisely the continuous-time mirror descent dynamics associated with $(-\mathcal{L})$, which is known to be a *gradient flow* in the spherical Hellinger–Kantorovich geometry (Mielke & Zhu, 2025). As such, $(-\mathcal{L})$ acts as a strict Lyapunov function for the system: along any non-stationary trajectory, $\frac{d}{dt}(-\mathcal{L})(\mathbf{Q}_t) < 0$. By (Benaïm, 2006, Corollary 6.6), every precompact APT converges to the set of stationary points of the Lyapunov function. Since the objective function $\mathcal{L}$ is the relative entropy, and hence strictly convex, its stationary point coincide with its global minimizer.

**Conclusion.** Combining (C1) and (C2) with Theorem E.2, we deduce that the interpolated process $\mathbf{h}(t)$ converges almost surely to the set of minimizers of $(-\mathcal{L})$, which readily implies that the original sequence $\{\mathbf{Q}^k\}$ inherits the same convergence guarantee. $\qquad\square$

# F DETAILED EXAMPLE OF ALGORITHM IMPLEMENTATION

In this section we provide comprehensive pseudocode of an example implementation for the two FINETUNINGSOLVER subprocedure in Alg. 4. It is implemented using a variation of Adjoint Matching (AM) which is introduced comprehensively in Domingo-Enrich et al. (2024), although we provide pseudocode below for completeness. We note that in principle one could substitute for any other linear fine-tuning method.

Before presenting the implementations, we shortly clarify some relevant notation. The algorithm makes explicit use of the interpolant schedules $\kappa_t$ and $\omega_t$ introduced in equation 1. We note that in flow model literature they are more commonly known as $\alpha_t$ and $\beta_t$. We denote by $u^{\text{pre}}$ the velocity field corresponding to the pre-trained policy $\pi^{\text{pre}}$, and likewise use $u^{\text{fine}}$ for the velocity field corresponding to the fine-tuned policy. In short, FINETUNINGSOLVER first samples trajectories, which are then used to approximate the solution of a surrogate ODE whose marginals are used as regression targets for the control policy (see Domingo-Enrich et al. (2024) Section 5 for a full discussion). We note that FINETUNINGSOLVER can be used for objectives with and without trajectory rewards, simply by setting trajectory rewards to zero.

---

**Algorithm 4** Adjoint Matching for fine-tuning Flow Matching models (FINETUNINGSOLVER)

---

**Require:** $u^{\text{pre}}$: pre-trained FM velocity field, $\{\nabla f_t\}_{t \in [0,1]}$: gradients of trajectory rewards, $\{\lambda_t\}_{t \in [0,1]}$: (optional) trajectory reward weights, $\gamma$: fine-tuning strength
1: Initialize fine-tuned vector fields: $u^{\text{finetune}} = u^{\text{pre}}$ with parameters $\theta$.
2: **for** $n \in \{0, \dots, N-1\}$ **do**
3:  Sample $m$ trajectories $\boldsymbol{X} = (X_t)_{t \in \{0,\dots,1\}}$ with memoryless noise schedule $\sigma(t) = \sqrt{2\kappa_t \left( \frac{\dot{\omega}_t}{\omega_t} \kappa_t - \dot{\kappa}_t \right)}$, e.g.:

$$X_{t+h} = X_t + h \left( 2u_\theta^{\text{finetune}}(X_t, t) - \frac{\dot{\omega}_t}{\omega_t} X_t \right) + \sqrt{h}\, \sigma(t)\, \varepsilon_t, \quad \varepsilon_t \sim \mathcal{N}(0, I), \quad X_0 \sim \mathcal{N}(0, I). \tag{51}$$

4:  For each trajectory, solve the *lean adjoint ODE* backwards in time from $t = 1$ to $0$, e.g.:

$$\tilde{a}_{t-h} = \tilde{a}_t + h\, \tilde{a}_t^\top \nabla_{X_t} \left( 2v^{\text{base}}(X_t, t) - \frac{\dot{\omega}_t}{\omega_t} X_t \right) - h\gamma\lambda_t \nabla_{X_t} f_t(X_t), \quad \tilde{a}_1 = \gamma\lambda_1 \nabla_{X_1} f_1(X_1). \tag{52}$$

5:  Note that $X_t$ and $\tilde{a}_t$ should be computed without gradients, i.e., $X_t = \texttt{stopgrad}(X_t)$, $\tilde{a}_t = \texttt{stopgrad}(\tilde{a}_t)$.
6:  For each trajectory, compute the following Adjoint Matching objective:

$$\mathcal{L}_{\text{Adj-Match}}(\theta) = \sum_{t \in \{0, \dots, 1-h\}} \left\| \tfrac{2}{\sigma(t)} \left( v_\theta^{\text{finetune}}(X_t, t) - u^{\text{base}}(X_t, t) \right) + \sigma(t)\, \tilde{a}_t \right\|^2. \tag{53}$$

7:  Compute the gradient $\nabla_\theta \mathcal{L}(\theta)$ and update $\theta$ using favorite gradient descent algorithm.
8: **end for**
**Output:** Fine-tuned vector field $v^{\text{finetune}}$

---

Crucially, we employ the fine-tuning oracle in Alg. 8 also to implement the projection step within Sec. 4, as indicated within Alg. 4. Moreover, in the case of a non-differentiable (weak or strong) verifier, the projection step can be implemented via a 0-th order RL-based fine-tuning method, e.g., (Fan et al., 2023; Black et al., 2023), which induce the same closed-form solution as Alg. 8.

## G    EXPERIMENTAL DETAILS

### G.1    ILLUSTRATIVE EXAMPLES EXPERIMENTAL DETAILS

Numerical values in all plots shown within Sec. 6 are means computed over diverse runs of FE via 5 different seeds. Error bars correspond to $95\%$ Confidence Intervals. For the following comparisons, we aimed to tune each algorithm parameters so that the method would work well in the specific illustrative example.

**Pre-trained models.** The pre-trained models appearing in Sec. 6, in the context of illustrative examples, are learned on synthetically generated data, via standard learning procedures. In particular, in Sec. 6 we always show samples generated by such pre-trained models.

**Global Flow Expansion.**

- For G-FE, we use $\lambda_t = 0$ if $t > 1 - 0.05$, and $\lambda_t = 1.2$ otherwise, $\gamma_k = \frac{1.5}{(1+3(k-1))}$, $\eta = 2$ and $K = 10$.

- For CONSTR we employ $\eta = 2$.

- For S-MEME we employ $\gamma_k = \frac{0.345}{(1+3(k-1))}$ and $K = 10$ and use $s_1^\pi(x) = s_{1-\epsilon}^\pi(x)$ with $\epsilon = 0.02$ as discussed in Sec. 4.

**Local Flow Expansion.** The models used act on a 2-dim state $(x_1, x_2)$, of which is shown only the $x_1$ coordinate in the process-level figures reported in Sec. 6. Since we use as oracle AM, which requires differentiable gradient, we consider a binary verifier (shown in Fig. 5c in grey), which we smoothen, thus rendering it differentiable and approximate. Notice that differentiability is not required by FE, but is rather an implementation detail due to the specific oracle used (i.e., AM (Domingo-Enrich et al., 2024), see Sec. F for further details). In particular, there exist several analogous oracles that do not require function differentiability (e.g., Fan et al., 2023).

- For L-FE we employ $K = 8$, $\lambda_t = 0$ if $t > 1 - 0.015$, and proportional with the process variance, i.e., $\lambda_t = \sqrt{2(\kappa_t(\frac{\dot{\omega}_t}{\omega_t}\kappa_t - \dot{\kappa}_t))}$, otherwise; $\gamma_k = 0.3$, $\eta_k = 0.1$.

- For FDC we use $K = 8$, $\gamma_k = 0.06$ and use $s_1^\pi(x) = s_{1-\epsilon}^\pi(x)$ with $\epsilon = 0.02$ as discussed in Sec. 4.

### G.2    CONFORMER VENDI

We begin with a detailed explanation of our diversity metric: conformer VENDI. In general, VENDI (Friedman & Dieng, 2022) is a diversity metric operating on arbitrary inputs based on a pairwise distance kernel $k : \mathcal{X} \times \mathcal{X} \to [0, 1]$. For a list of inputs $x_1, \ldots, x_n$ and a symmetric pairwise distance kernel $k$, VENDI is defined as:

$$VS(x_1, \ldots, x_n) = \exp\left(-\sum_{i=1}^{n} \lambda_i \log \lambda_i\right) \tag{52}$$

where $\lambda_i$ are eigenvalues of the distance matrix $K$ with $K_{ij} = k(x_i, x_j)$. In our work, we use a molecular fingerprinting method combined with a kernel simply defined as the Euclidean distance between fingerprints, thereby inducing a kernel on molecules themselves. The particular fingerprinting method is defined as the sorted list of pairwise atomic distances. Formally, for a molecule with $N$ atoms at positions $a_1, \ldots a_N$ we first compute the matrix of pairwise distances $A_{ij} = \|a_i - a_j\|_2$, $1 \le i < j \le N$, which is then sorted as $\tilde{A}_{i_1 j_1} \ldots \tilde{A}_{i_{N(N-1)/2} j_{N(N-1)/2}}$ yielding the fingerprint $\tilde{A} \in \mathbb{R}^{N(N-1)/2}$.

### G.3    PCA PROJECTION FOR FIG. 5D

In this section we explain the dimensionality reduction method used to generate the plot in 5d. We first generated 25000 molecules from both the pre-trained model $\pi^{\text{pre}}$ (yielding $D^{\text{pre}}$ and the fine-tuned

model $\pi^K$ (yielding $D^{\text{finetuned}}$), which was computed by the L-FE algorithm on $\pi^{\text{pre}}$ for $K = 5$ iterations with $\alpha = 9$. We then fingerprinted each set of molecules using the method described above, and fit a 1-dim PCA on the fingerprints for $D^{\text{pre}}$ using SCIKIT-LEARN (Pedregosa et al., 2011), which was then used to transform both $D^{\text{pre}}$ and $D^{\text{finetuned}}$ into 1-dimensional vectors. Fig. 5d corresponds to a histogram plot of each of the resulting sets of vectors.

### G.4  VALIDITY COMPUTATION IN MOLECULAR EXPERIMENTS

In the context of our experiments on molecules the concept of validity is defined through a pipeline of several checks, defined below. Our validity function passes through each one sequentially, returning an invalid result if any fail, and a valid result only if all checks pass.

1. First, we attempt to sanitize each molecule using RDKit's (RDKit) CHEM.SANITIZEMOL function. As an added check, we test if it is possible to convert the molecule to and back from SMILES (Weininger, 1988) notation.

2. We then iterate over each atom in the molecule, checking for any implicit hydrogens (our model must generate explicit hydrogens as FlowMol (Dunn & Koes, 2024) does) or any radical electrons which would make the molecule invalid.

3. Finally we perform our weak verifier check, filtering out molecules for which any two atoms are closer than 0.9Å. Details on this weak verifier are explained below.

The final validity check is evaluate the weak verifier on molecules that pass the previous steps. The weak verifier itself is evaluated by first computing the vector of pairwise distances between atoms $\tilde{A}$ (see discussion in Section G.2 above), then taking the minimum element $\tilde{A}_0$ and checking if it is lower than 0.9Å, in which case a molecule is classified as invalid. Including this check in the validity function guarantees by construction that the weak verifier satisfies Definition 2, since failing the weak verifier check implies failing the validity check as well.

### G.5  PRACTICAL DETAILS FOR EXPERIMENTS ON MOLECULES

In this section we discuss the practical choices behind the molecular design experiments discussed in Section 6. We start with a discussion of hyperparameter settings, followed by some implementation techniques adapting the verifier feedback for a first-order solver, and finally discuss hardware and platform used for training.

#### G.5.1  HYPERPARAMETER CHOICES

For our experiments on molecules we use the FlowMol CTMC model from Dunn & Koes (2024) trained on the QM9 dataset as a pre-trained model. We run each algorithm (L-FE and FDC) with the following parameters:

- $K = 5$ iterations
- Regularization strength of $\alpha = 9$
- Decreasing stepsize of $\gamma_k = \frac{\gamma_0}{1+k}$ with $\gamma_0 = 0.00001$
- For the trajectory reward weighting (L-FE only) we use $\lambda_t = \sigma_t = \sqrt{2\kappa_t\left(\frac{\dot{\omega}}{\omega}\kappa_t - \dot{\kappa}_t\right)}$, ensuring $\lambda_t \to 0$ as $t \to 1$ for stability as discussed at the end of section 4
- For both, we clip the score near the end of the trajectory as $s_t^\pi = s_{\{\min t, 1-\epsilon\}}^\pi$ for $\epsilon = 0.005$.
- We fix the number of atoms in generated molecules (for model training and metric calculation) to 10, in order to simplify metric calculations.

When using Adjoint Matching (AM) (Domingo-Enrich et al., 2024) to implement the subroutines of any algorithm we use $N = 4$ iterations, we sample a batch of $m = 4$ trajectories of length 40 at each iteration and update the parameters $\theta$ using Adam Kingma & Ba (2015) with a learning rate of 0.00055. We note that since FlowMol CTMC is a mixed categorical and continuous flow model, we only use AM to update the parameters corresponding to the continuous outputs of the model, i.e., the atom positions.

### G.5.2 Hyperparameter Choices for Experiments on GEOM

In order to test the performance of our model in a more practical setting, we performed additional experiments using the GEOM dataset. We again use a FlowMol CTMC model from Dunn & Koes (2024) as a pre-trained model, however this time using checkpoints from training on the GEOM dataset. Note that for simpler hyperparameter search we use the reparametrization discussed in section H. The optimal parameter set for each algorithm (L-FE, G-FE, NSE, and FDC) is not identical in this setting, therefore we first report the hyperparameters in common before listing the differences for each algorithm below:

**Common hyperparameters for L-FE, G-FE, NSE, FDC:**

- $K = 3$ iterations
- Constant (adjusted) stepsize $\tilde{\gamma}_k = \tilde{\gamma}_0$, although the magnitude $\tilde{\gamma}_0$ differs for each algorithm
- For the trajectory reward weighting (L-FE, G-FE and NSE) we use $\lambda_t = \sigma_t = \sqrt{2\kappa_t\left(\frac{\dot{\omega}}{\omega}\kappa_t - \dot{\kappa}_t\right)}$, ensuring $\lambda_t \to 0$ as $t \to 1$ for stability as discussed at the end of section 4
- For all methods, we clip the score near the end of the trajectory as $s_t^\pi = s_{\min\{t,1-\epsilon\}}^\pi$ for $\epsilon = 0.005$.
- We fix the number of atoms in generated molecules (for model training and metric calculation) to 30, in order to simplify metric calculations.

**Per-algorithm hyperparameter variations:**

- L-FE:
    - $\beta = 0.4$
    - $\tilde{\gamma}_k = 0.0005$ (constant)
    - $\eta_k = 5.0$ (constant)
- NSE:
    - $\beta = 0.0$
    - $\tilde{\gamma}_k = 0.0002$ (constant)
- FDC:
    - $\beta = 0.9$
    - $\tilde{\gamma}_k = 0.0005$ (constant)

When using Adjoint Matching (AM) (Domingo-Enrich et al., 2024) to implement the subroutines of any algorithm usign the GEOM FlowMol model we use $N = 4$ iterations, we sample a batch of $m = 1$ trajectories of length 40 at each iteration and update the parameters $\theta$ using Adam Kingma & Ba (2015), with a learning rate of 0.0001. We note that since FlowMol CTMC is a mixed categorical and continuous flow model, we only use AM to update the parameters corresponding to the continuous outputs of the model, i.e., the atom positions.

### G.5.3 Smoothing the Weak Verifier

Since we use Adjoint Matching for all fine-tuning tasks we need all rewards to be differentiable. While our weak verifier is formally defined as $v(x) = 1 \iff x$ respects the minimum atom separation bound of 0.9Å, we use the following differentiable approximation using a sigmoid soft indicator function:

$$v(x) = \frac{1}{N(N-1)/2} \sum_{i=1}^{N(N-1)/2} \frac{\exp(\tilde{A}_i - 0.9)}{\exp(\tilde{A}_i - 0.9) + 1} \tag{53}$$

where $\tilde{A}_i$ are the pairwise atomic distances introduced in Section G.2 above. This alternative verifier is differentiable and provides gradient feedback everywhere and therefore can be used in Adjoint Matching.

### G.5.4 ABLATION STUDY

We report an ablation study of the key hyperparameters $\alpha$, $\gamma_k$, and $\eta_k$. Notice first that since L-FE is a generalisation of both NSE and G-FE, we recover NSE by setting $\eta_k = 0$, and recover G-FE by instead setting $\alpha = 0$. The following plot shows a comparison of the Pareto fronts for each method, all compared against FDC. Furthermore, at the end of this section, we report results of running the projection step alone (effectively setting $\gamma_k = 0$).

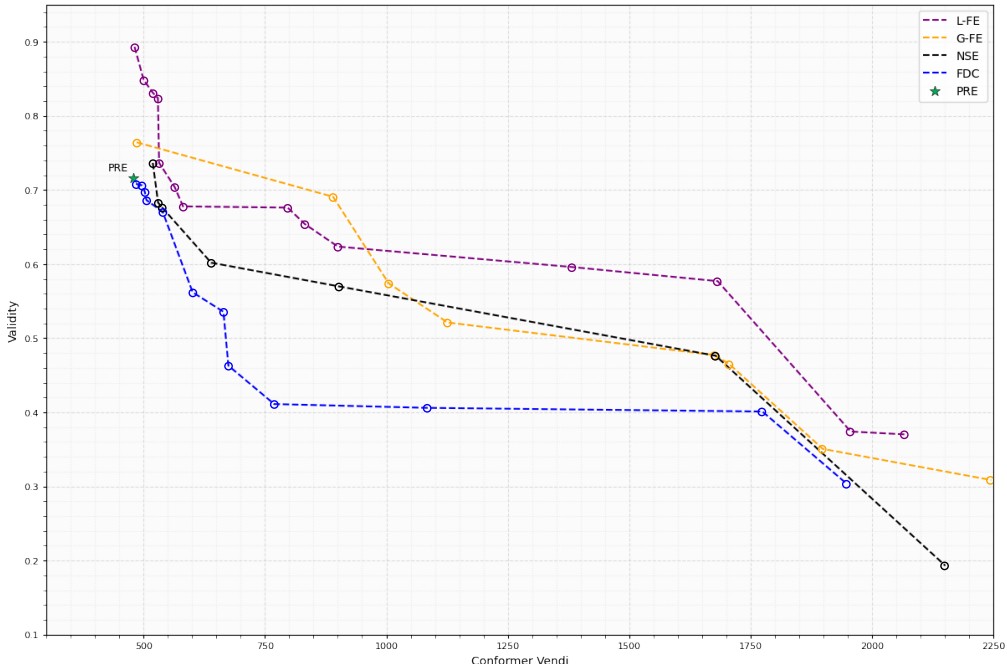

Figure 7: Comparison of different parametrization of L-FE, G-FE, NSE, FDC, for $K = 3$. We compute Conformer Vendi and validity over 5000 samples, and average results over 8 runs for each parameterization with 8 different seeds.

In Fig. 7 we report all the Pareto dominant points chosen for each method evaluated in the following ranges of hyperparameters (using the reparametrized notation from Appendix H):

- $\beta$: 0.0, 0.1, 0.2, 0.3, 0.4, 0.5, 0.6, 0.7, 0.8, 0.9, 0.95

- $\tilde{\gamma}_k$ (constant): 0.0001, 0.0002, 0.0003, 0.0004, 0.0005, 0.001

- $\eta_k$ (constant): 0.0, 0.1, 0.5, 1.0, 2.0, 5.0

For each combination of the hyperparameters above, we run $K = 3$ iterations of each method (L-FE, G-FE and NSE) and average the results across 8 seeds. Conformer Vendi and validity are always computed for batches of 5000 samples. For all other hyperparameters refer to the previous section G.5.2 discussing general hyperparameter choices for the GEOM model. To build the plot above, we simply drop all points that are Pareto dominated (i.e. same/greater Conformer Vendi and same/greater validity) by some other point, and plot the remaining points. We also drop all points with Conformer Vendi lower than the pre-trained model. The results shown in Fig. 7 clearly demonstrate the ability of each method to trade off validity for higher diversity (measured by Conformer Vendi). However notice that only G-FE and L-FE can significantly increase validity due to their use of verifier information. Overall, notice that both L-FE and G-FE outperform all other methods almost uniformly, and NSE uniformly outperforms FDC (both unconstrained exploration algorithms). L-FE is especially effective at retaining or even increasing validity: it significantly dominates all other methods in terms of validity in the range of 475 to 550 Conformer Vendi (top left of Fig. 7).

### G.5.5 TABLES OF RESULTS

We report the numerical results (value and confidence interval) for each point in Fig. 7 in the tables below. For each method we report the mean and confidence interval corresponding to each point from left to right, and report the parameterization ($\beta$, $\tilde{\gamma}_0$ and $\eta$ if applicable) used to generate the point. We acknowledge the size of the confidence intervals increases drastically in the regime of more exploratory points (higher Conformer Vendi): increasing stability of exploration methods in that regime remains an open problem. Still, we note that L-FE dominates other methods most significantly when the Conformer Vendi increase is modest (less than $550$ Conformer Vendi), and in that regime the confidence intervals are reasonably concentrated.

Table 1: Values for L-FE reported in Fig 7 with 95% confidence intervals

| $\beta$ | $\tilde{\gamma}_0$ | $\eta_0$ | Mean Validity (95% CI) | Mean Conformer Vendi (95% CI) |
|---|---|---|---|---|
| 0.3 | 0.0002 | 5.0 | 0.89 (0.87 $\pm$ 0.91) | 481.35 (460.24 $\pm$ 500.42) |
| 0.6 | 0.0005 | 5.0 | 0.85 (0.79 $\pm$ 0.89) | 500.11 (460.72 $\pm$ 546.91) |
| 0.4 | 0.0005 | 5.0 | 0.83 (0.76 $\pm$ 0.89) | 519.05 (454.45 $\pm$ 585.99) |
| 0.1 | 0.0004 | 5.0 | 0.82 (0.70 $\pm$ 0.90) | 529.47 (465.92 $\pm$ 629.29) |
| 0.2 | 0.0004 | 5.0 | 0.74 (0.53 $\pm$ 0.88) | 531.90 (452.14 $\pm$ 633.92) |
| 0.2 | 0.0005 | 5.0 | 0.70 (0.53 $\pm$ 0.86) | 563.75 (472.19 $\pm$ 683.79) |
| 0.2 | 0.0004 | 2.0 | 0.68 (0.50 $\pm$ 0.83) | 581.26 (467.44 $\pm$ 728.66) |
| 0.3 | 0.0005 | 5.0 | 0.68 (0.46 $\pm$ 0.85) | 796.64 (475.55 $\pm$ 1371.21) |
| 0.2 | 0.0005 | 2.0 | 0.65 (0.43 $\pm$ 0.84) | 831.91 (496.49 $\pm$ 1423.91) |
| 0.1 | 0.0004 | 2.0 | 0.62 (0.39 $\pm$ 0.82) | 900.08 (515.31 $\pm$ 1591.61) |
| 0.1 | 0.0005 | 5.0 | 0.60 (0.33 $\pm$ 0.83) | 1382.46 (516.37 $\pm$ 2493.28) |
| 0.4 | 0.0005 | 0.5 | 0.58 (0.31 $\pm$ 0.81) | 1682.32 (560.86 $\pm$ 3320.25) |
| 0.1 | 0.0005 | 2.0 | 0.37 (0.13 $\pm$ 0.64) | 1956.33 (684.10 $\pm$ 3322.31) |
| 0.1 | 0.0005 | 0.5 | 0.37 (0.12 $\pm$ 0.64) | 2068.02 (866.88 $\pm$ 3366.56) |

Values are mean (95% confidence interval). Confidence intervals computed using bootstrapping and shown to two decimal places.

Table 2: Values for G-FE reported in Fig 7 with 95% confidence intervals

| $\tilde{\gamma}_0$ | $\eta_0$ | Mean Validity (95% CI) | Mean Conformer Vendi (95% CI) |
|---|---|---|---|
| 0.0002 | 0.5 | 0.76 (0.69 $\pm$ 0.83) | 485.65 (442.52 $\pm$ 524.71) |
| 0.0005 | 5.0 | 0.69 (0.46 $\pm$ 0.86) | 890.19 (434.36 $\pm$ 1731.52) |
| 0.0004 | 0.5 | 0.57 (0.33 $\pm$ 0.79) | 1004.79 (531.74 $\pm$ 1790.78) |
| 0.0004 | 1.0 | 0.52 (0.27 $\pm$ 0.77) | 1125.94 (521.23 $\pm$ 2182.77) |
| 0.0004 | 0.1 | 0.48 (0.22 $\pm$ 0.72) | 1676.67 (533.47 $\pm$ 2888.04) |
| 0.0005 | 1.0 | 0.46 (0.22 $\pm$ 0.71) | 1706.53 (664.86 $\pm$ 3100.05) |
| 0.0005 | 2.0 | 0.35 (0.12 $\pm$ 0.60) | 1898.72 (796.05 $\pm$ 3221.32) |
| 0.0005 | 0.5 | 0.31 (0.08 $\pm$ 0.56) | 2245.62 (1135.37 $\pm$ 3500.43) |

Values are mean (95% confidence interval). Confidence intervals computed using bootstrapping and shown to two decimal places.

Table 3: Values for NSE reported in Fig 7 with 95% confidence intervals

| $\beta$ | $\tilde{\gamma}_0$ | Mean Validity (95% CI) | Mean Conformer Vendi (95% CI) |
|---|---|---|---|
| 0.0 | 0.0002 | 0.74 (0.68 $\pm$ 0.81) | 518.92 (505.70 $\pm$ 531.90) |
| 0.4 | 0.0004 | 0.68 (0.58 $\pm$ 0.77) | 529.85 (484.34 $\pm$ 580.24) |
| 0.3 | 0.0004 | 0.68 (0.55 $\pm$ 0.79) | 537.83 (491.25 $\pm$ 588.98) |
| 0.1 | 0.0004 | 0.60 (0.42 $\pm$ 0.77) | 639.04 (501.23 $\pm$ 857.00) |
| 0.2 | 0.0004 | 0.57 (0.37 $\pm$ 0.74) | 902.23 (534.29 $\pm$ 1559.95) |
| 0.1 | 0.0005 | 0.48 (0.25 $\pm$ 0.69) | 1678.29 (669.60 $\pm$ 3024.30) |
| 0.0 | 0.0005 | 0.19 (0.02 $\pm$ 0.40) | 2151.14 (1256.65 $\pm$ 3162.47) |

Values are mean (95% confidence interval). Confidence intervals computed using bootstrapping and shown to two decimal places.

Table 4: Values for FDC reported in Fig 7 with 95% confidence intervals

| $\beta$ | $\tilde{\gamma}_0$ | Mean Validity (95% CI) | Mean Conformer Vendi (95% CI) |
|---|---|---|---|
| 0.4 | 0.0001 | 0.71 (0.67 $\pm$ 0.74) | 484.34 (461.47 $\pm$ 506.75) |
| 0.5 | 0.0002 | 0.71 (0.63 $\pm$ 0.77) | 496.06 (467.29 $\pm$ 523.97) |
| 0.7 | 0.0002 | 0.70 (0.64 $\pm$ 0.75) | 502.49 (483.86 $\pm$ 526.78) |
| 0.5 | 0.0001 | 0.69 (0.65 $\pm$ 0.74) | 505.84 (468.66 $\pm$ 538.20) |
| 0.2 | 0.0003 | 0.67 (0.58 $\pm$ 0.77) | 539.29 (504.70 $\pm$ 585.65) |
| 0.9 | 0.001 | 0.56 (0.49 $\pm$ 0.62) | 601.06 (531.19 $\pm$ 699.07) |
| 0.3 | 0.0004 | 0.54 (0.34 $\pm$ 0.72) | 664.45 (531.77 $\pm$ 847.98) |
| 0.5 | 0.0005 | 0.46 (0.30 $\pm$ 0.66) | 674.63 (544.65 $\pm$ 824.76) |
| 0.4 | 0.0005 | 0.41 (0.25 $\pm$ 0.58) | 768.98 (599.68 $\pm$ 975.95) |
| 0.1 | 0.0004 | 0.41 (0.24 $\pm$ 0.56) | 1084.21 (574.88 $\pm$ 2062.43) |
| 0.1 | 0.0005 | 0.40 (0.15 $\pm$ 0.66) | 1774.38 (828.37 $\pm$ 2904.44) |
| 0.2 | 0.0005 | 0.30 (0.07 $\pm$ 0.56) | 1948.54 (954.87 $\pm$ 3004.34) |

Values are mean (95% confidence interval). Confidence intervals computed using bootstrapping and shown to two decimal places.

### G.5.6 Ablation: pure EXPAND and PROJECT steps

In order to isolate the effects of the EXPAND and PROJECT steps, we report the result of running each for $K = 3$ iterations in Fig. 8. Fig. 8 shows the Pareto optimal points of the following hyperparameter combinations:

EXPAND step (NSE, black circles in Fig. 8)

- $\beta$: 0.0, 0.1, 0.2, 0.3, 0.4, 0.5, 0.6, 0.7, 0.8, 0.9, 0.95
- $\tilde{\gamma}_0$: 0.0001, 0.0002, 0.0003, 0.0004, 0.0005

PROJECT step (green circles in Fig. 8)

- $\eta_k = \eta_0$ constant set to: 0.0, 0.1, 0.5, 1.0, 2.0, 5.0

We remark again that running only the EXPAND step without the PROJECT step is equivalent to running NSE. For each parameter combination above we average results over 8 seeds. Notice that the EXPAND and PROJECT steps have the reverse effect of each other: the EXPAND step trades off validity for increased diversity (Conformer Vendi) where as the PROJECT step increases validity at a cost of marginally reduced diversity.

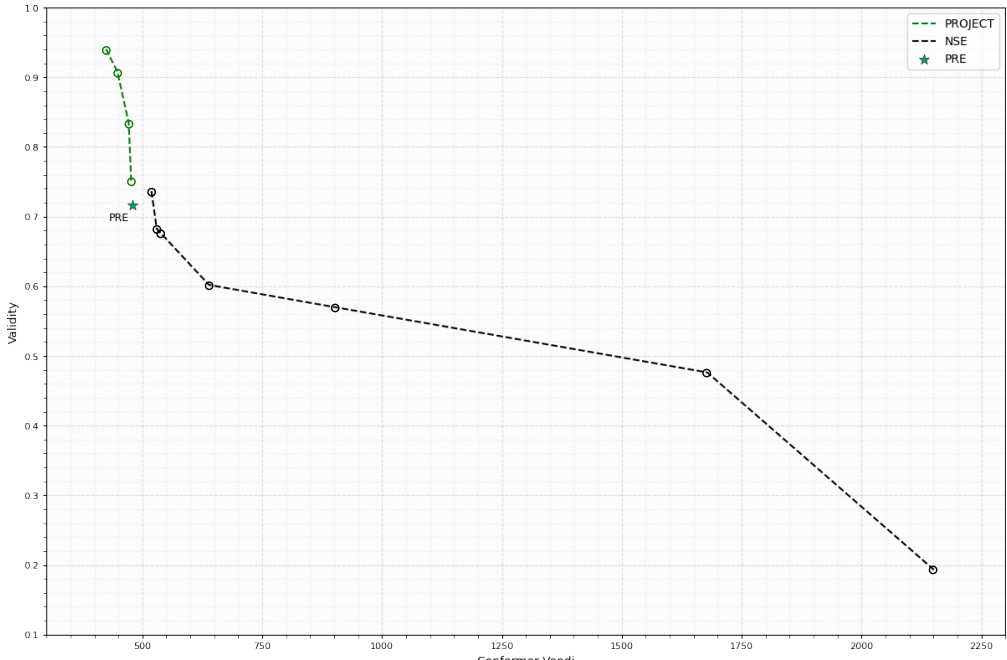

Figure 8: Illustrating the result of isolating the EXPAND step (NSE) and the PROJECT step

### G.5.7 COMPARISON WITH INFERENCE TIME FILTERING

In this section, we consider an additional comparison of the proposed methods against a baseline corresponding to FDC plus inference-time filtering. We report the results in Fig. 9 below. In particular, we consider the following algorithms:

- FDC (De Santi et al., 2025b) + inference-time filtering (blue in Fig. 9)
- NSE (Alg. 3 + inference-time filtering (black in Fig. 9)
- L-FE

In particular, for the first two cases above, given a model fine-tuned using FDC or NSE, we filter the samples generated at inference time by using the same weak verifier employed by L-FE. We note that this schemes induce a closed-form solution formally mathematically equivalent to conditional sampling, with the condition that the samples are in $\Omega_v$. Thus, this schemes effectively amounts to simulating a perfect projection step, albeit at the price of costly per-sample inference-time filtering (e.g.,, via rejection sampling). We calculate the Conformer Vendi and validity on each set of 5000 molecules, and report the results in Fig. 9 below. As one can notice, NSE + filtering shows superior performance compared against FDC + filtering, while L-FE shows slightly less validity than NSE + filtering, while being significantly cheaper at sampling time, which is a needed requirement for certain generative modeling applications. Compared with FDC + filtering, L-FE shows superior exploration capabilities and slightly lower validity. Nonetheless, it might be possible to parametrize L-FE to achieve lower diversity and higher validity, similarly to FDC + filtering. In particular, these two algorithms theoretically have the same closed-form solutions, and the concrete differences amount to the high per-sample cost of inference-time filtering, and the superior exploration capabilities of L-FE and NSE over FDC likely due to noised space exploration, as discussed in Sec. 2.

Each point in Fig. 9 corresponds to a different Pareto optimal parametrization ($\beta$, $\gamma$ and $\eta$ if applicable) for each method. Each method is run for $K = 3$ iterations, with results averaged across 8 seeds. See section G.5.2 for details about other hyperparameters.

### G.5.8 HARDWARE

We ran all of our experiments using a single NVIDIA RTX 2080Ti GPU per run (QM9 experiments) or a single NVIDIA RTX 4090 GPU per run (GEOM experiments).

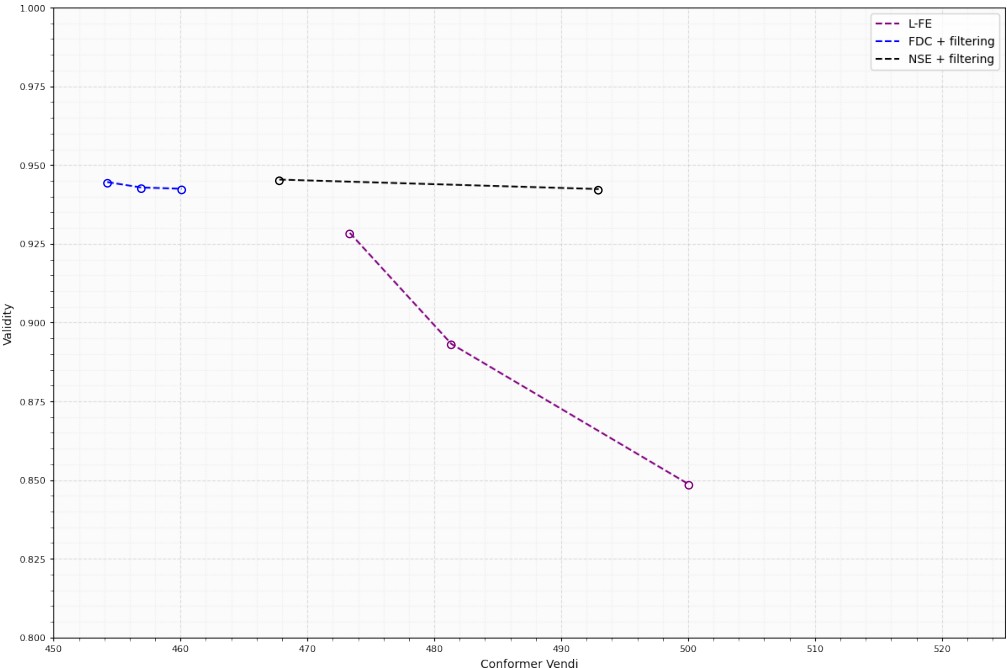

Figure 9: L-FE compared to NSE and FDC with inference-time filtering for different parameterizations. We note that L-FE maintains competitive performance with both methods despite having no filter applied to the output samples, and NSE still outperforms FDC when both are corrected with an inference-time filter.

## H UPDATE STEP REPARAMETRIZATION

Recall the expression for the gradient of running costs in the Local Flow Expander algorithm (Alg. 2, equation 14):

$$\lambda_t \nabla_{x_t} \delta \mathcal{G}(p_t^\pi) = \lambda_t \nabla_{x_t} \delta(\mathcal{H}(p_t^\pi) - \alpha_t \mathcal{D}_{\text{KL}}(p_t^\pi || p_t^{\text{pre}})) \tag{54}$$

$$= \lambda_t(-s_t^\pi + \alpha(s_t^\pi - s_t^{\text{pre}})) \tag{55}$$

$$= -\lambda_t((\alpha + 1)s_t^\pi - \alpha s_t^{\text{pre}}) \tag{56}$$

which are then multiplied by the stepsize $\gamma_k$ at each iteration, resulting in the following expression being plugged into the Adjoint Matching algorithm as the gradient of the running cost:

$$\nabla f_t = -\gamma_k \lambda_t((\alpha + 1)s_t^\pi - \alpha s_t^{\text{pre}}). \tag{57}$$

While $\alpha$ has an intuitive interpretation as the regularization strength in objective 7, it has the unfortunate side-effect of scaling the magnitude of the running cost which could potentially have the opposite effect. Indeed, notice that as $\alpha \to \infty$ the running costs explode. For practical applications it seems more suitable to reparametrize the running cost as follows:

$$\nabla f_t = -\tilde{\gamma}_k \lambda_t(s_t^\pi - \beta s_t^{\text{pre}}) \tag{58}$$

for $\beta = \frac{\alpha}{\alpha+1} \in [0,1]$, absorbing a $(\alpha + 1)$ factor into the new stepsize $\tilde{\gamma}_k$:

$$\tilde{\gamma}_k = (\alpha + 1)\gamma_k. \tag{59}$$

Note that this parametrization is as expressive as before but is easier to tune as it disentangles the effect of the stepsize and the regularization strength.

