# OpenReview forum: "Verifier-Constrained Flow Expansion for Discovery Beyond the Data"
_ICLR.cc/2026/Conference — ICLR 2026 Poster_

### Official Review · Reviewer_e76w · 2025-10-31

**Soundness:** 2
**Presentation:** 2
**Contribution:** 2
**Rating:** 6
**Confidence:** 3

**Summary:**

This paper proposes Flow Expander (FE), a general framework for expanding the support of pre-trained flow models to cover the valid design space more uniformly. The method uses verifiers—functions that check the validity of samples (e.g., physical or chemical constraints)—to guide exploration and improve the generative model’s coverage beyond its initially narrow region.

**Strengths:**

- Introduces a method to expand flow models using strong or weak verifiers to expand diversity.
- The scheme allows practical, gradient-based fine-tuning compatible with existing flow and diffusion models.

**Weaknesses:**

- What is $\lambda_t$ and $\gamma_t$ in Eq. 8 and 9? Are these weightings and discount factors? And how are they chosen?
- How do you parametrize the verifier function $v(x)$, is it modeled like a Gumbel-Softmax? Also, I wonder if it's strict to have $v(x)$ to be bounded, is the method extendable if one has an unbounded $v(x)$?
- I am not sure about the downstream applications of this method in drug discovery, as currently it's only restricted to the bounded verifier objective to check validity or bonds. Usually, this can also be done via BO in latent space by embedding priors with GP or just generally with rejection sampling in latent space to get valid molecules only, similar to an explore and exploit-based method. If the $v(x)$ can be extended to the unbounded domain such that one can use molecular properties to guide it, then it would be a good contribution.
- Does the model incorporate whether the verifier is noisy?
- Could FE be viewed as iteratively reshaping an implicit energy landscape defined by verifier penalties?

**Questions:**

See the weaknesses.

---

> ### Author Response · Authors · 2025-11-25
>
> We thank the Reviewer for recognizing our method as practical and compatible with existing models. In the following, we aim to sharply clarify the points raised by the Reviewer.
>
> **$\lambda_t$, $\gamma_t$, and $\alpha_t$**
>
> Since a parameter $\gamma_t$ does not exist, the question is ambiguous, and to be on the safer side, in the following we report brief explanations for both $\alpha_t$ (in Eq. 9) and $\gamma_k$ (in Eq. 10).
>
> 1. $\lambda_t$ is a parameter that allows to weight the objective $\mathcal{G}_t$ (e.g., entropy) non-uniformly over the time interval of the integral in Eq. 8. For instance, for $\mathcal{G} = \mathcal{H}$ corresponding to the entropy functional, $\lambda_t$ renders it possible to prioritize exploration in different time intervals of the flow process, and allows to stabilize the proposed method by choosing $\lambda_t = 0 $ for $t \to 1$, as discussed in lines 317-318. This parametrization for $\lambda_t$ solves the score explosion issue presented in Eq. 19 for $t \to 1$, thus rendering the method more stable, especially in higher dimensional settings, as shown in Fig. 6.b.
>
> 2. $\gamma_k$ in Eq. 9 is a standard step-size in a mirror descent, or proximal gradient descent, scheme. Effectively, Eq. 8 describes an optimization problem over a probability space of flow processes. As a consequence, $\gamma_k$ controls how close (in terms of KL-divergence) a new iterate (i.e. a flow process $Q^k$) should be in probability space from the current flow process $Q^{k-1}$. Intuitively, higher values of $\gamma_k$ lead to weaker regularization between iterates, while lower values to stronger regularization and therefore smoother paths in probability space. Concretely, $\gamma_k$ has to be chosen experimentally as it depends on the complex landscape of the optimization problem, as it is the case for the vast majority of deep learning optimization schemes.
>
> 3. $\alpha_t$ is a straightforward time-varying extension of the parameter $\alpha$ introduced and discussed in Sec. 3 (lines 207-216). In short, it controls the amount of prior-regularization strength (in terms of KL-divergence, see Eq. 7) of the optimal solution. Lower values of $\alpha$ lead to a more expanded flow model, while higher values to a more conservative form of expansion. While the previously discussed $\gamma_k$ controls the rate of convergence of the method, $\alpha_t$ controls the optimal (asymptotic) solution of the method. In other words, $\alpha_t$ is part of the optimization objective (see Eq. 8 and 9), while $\gamma_k$ is purely part of the algorithmic scheme that we employ. Reasonable values of $\alpha_t$ require to be experimentally tuned. Nonetheless, one can use sufficient prior judgment via the mentioned optimization viewpoint to make an educated guess for both values.

---

> > ### Author Response · Authors · 2025-11-25
> >
> > **Verifier Structure**
> >
> > The true verifier is currently modeled as a function $v: \mathcal{X} \to \{0,1\}$ (i.e., a binary function). In practice, the (weak or strong) verifier leveraged by our method (FE), can either be a differentiable approximation, as it is the case for the experiments in Sec. 4, as discussed in Appendix G.5.3, or it can be a non-differentiable feedback oracle. While certain available verifiers are given by differentiable (learned) neural approximators or direct differentiable computations (e.g., interatomic distances checkers), this is not always the case. In fact, many real-world verifiers are non-differentiable. Crucially, our method (FE, Alg. 2) is compatible with both differentiable and non-differentiable verifiers. Notice that the only change dictated by the type of verifier regards the choice of the FineTuningSolver oracle for the projection step (line 3, Alg. 1). In particular, there exist a wide variety of FineTuningSolver oracles for diffusion and flow models that are compatible with differentiable functions (e.g., Adjoint Matching [1], which we employ in Sec. 6), as well as typical RL-based fine-tuning schemes, which are compatible with 0-th order information (i.e., function evaluations) and therefore do not require differentiability of the verifier (see e.g., [2,3]), or even not RL-based (e.g., [4]).
> >
> > We are not sure what the Reviewer means exactly by 'unbounded verifier'. If possible, we kindly ask the Reviewer to further clarify (e.g., mathematically) what they mean by 'unbounded' in this context. If they mean that it is a real function $v: \mathcal{X} \to R$, with an unbounded image (e.g., the real numbers), then our formal problem and method can be straightforwardly extended to such case at least in two ways:
> >
> > 1. By thresholding the verifier and hence rendering it binary. Or,
> >
> > 2. By interpreting the verifier of the form $v: \mathcal{X} \to R$ as a typical reward function for reward-guided adaptation of diffusion/flow models (see e.g., [1]) and extend our formulation to one of reward-guided expansion. This is immediate as it simply requires to add a linear (functional) term to Eq. 8, and to the best of our understanding the vast majority of the contributions (e.g., method logic, guarantees etc.) would trivially extend to this setting.
> >
> > Please, let us know if the Reviewer was referencing to another interpretation of boundedness/unboundedness of the verifier. We would be very happy to further clarify this point!
> >
> > **Guidance via Molecular Properties**
> >
> > To the best of our understanding of the Reviewer observation, they are referring to a form of property-guided expansion process. In this case, this is certainly achievable by a straightforward extension of our approach, simply by considering a reward function of the form $v: \mathcal{X} \to R$ additively added to the entropic term in Eq. 5 and 6 (and 8), as pointed out within point (2) of the previous paragraphs. The presented algorithmic scheme, as well as the rest of the framework, would be directly applicable to this property-guided expansion problem with minimal changes.
> >
> > **Noisy Verifier**
> >
> > In this work, we assume that the verifier is noise-free. In fact, in our experiments the RDKit pipeline corresponds to a deterministic rule-based checker, so no additional modeling of verifier noise is needed. However, the presented Flow Expansion framework and method can be naturally extended to noisy verifiers. Concretely, it might be sufficient to model the expectation of a noisy binary oracle as a soft verifier (i.e., acceptance probability) and concentrate its estimation by multiple evaluations. Nonetheless, this approach would offer only a first practical approach to tackle the noisy-verifier setting. We believe that a more principled viewpoint on this problem would be provided by a bandit (or Bayesian optimization) formulation, according to which diverse modeling of the noise structure would lead to diverse 'active' strategies to effectively solve the exploration-exploitation problem. Crucially, the current work assumes that the verifier is known, and tackles only the optimization problem of flow expansion. We believe that a setting where the verifier is unknown, and therefore one wishes to tackle the statistical problem of exploration-exploitation would be a highly-relevant extension of this work.

---

> > > ### Author Response · Authors · 2025-11-25
> > >
> > > **FE as iterative energy landscape reshaping**
> > >
> > > In a certain, intuitive and approximate sense, yes. In particular, standard entropy-regularized fine-tuning of diffusion/flow models (i.e., our FineTuningSolver oracle) aims to sample according to a density which trades-off an energy term given by a reward function, and the generative prior. This can be expressed via a closed form solution showing the two terms controlled by the prior-regularization strength $\alpha$ (see e.g., Eq. 1 in [1]). Our algorithm (FE, Alg. 2) effectively performs sequential fine-tuning of a pre-trained diffusion/flow model via a FineTuningSolver oracle with surrogate rewards automatically determined by the first variation of the functional $\mathcal{G}_t$. As a consequence, FE effectively iteratively adapts the prior density to approximate an optimal density determined by the energy landscape induced by the verifier, as well as the entropy term to promote expansion. Interestingly, also this density can be described in closed-form, and for the case of global expansion it corresponds to the uniform density over the valid set induced by a verifier, as discussed in Sec. 3.1. Ultimately, we wish to point out that while imagining the process as "iteratively reshaping an implicit energy landscape" might bring useful intuition, this is formally not correct since energy functions are induced by linear functionals (e.g., reward/verifier), but the entropy term is non-linear and to our knowledge the classic notion of energy does not apply. Nonetheless, one can still derive a very intuitive closed-form solution, as previously discussed and explained in Sec. 3.1 for the problem of global expansion.
> > >
> > > **References**
> > >
> > > [1] Adjoint matching: Fine-tuning flow and diffusion generative models with memoryless stochastic optimal control, C. Domingo-Enrich, 2024.
> > >
> > > [2] Training diffusion models with reinforcement learning, K. Black, 2023.
> > >
> > > [3] Dpok: Reinforcement learning for fine-tuning text-to-image diffusion models, Y. Fan., 2023.
> > >
> > > [4] Diffusion model alignment using direct preference optimization, B. Wallace, 2024.

---

### Official Review · Reviewer_MbXc · 2025-10-31

**Soundness:** 2
**Presentation:** 3
**Contribution:** 1
**Rating:** 2
**Confidence:** 4

**Summary:**

This paper addresses the problem that pre-trained generative models (flows, diffusion) tend to sample from a narrow part of the valid design space, which is a key limitation for scientific discovery. The authors propose to "expand" the model's density by leveraging an external verifier. The paper formalizes this into two problems: Global Flow Expansion (using a perfect strong verifier) and Local Flow Expansion (using an imperfect weak verifier).

To solve these, the paper introduces Flow Expander (FE), a mirror descent algorithm. A key idea is to formulate the optimization objective over the entire noised state space of the flow process ($Q^\pi = \{p_t^\pi\}_{t \in [0,1]}$) rather than just the final time-step $p_1^\pi$. This is claimed to provide a principled way to avoid score divergence issues near $t=1$. The paper provides theoretical convergence guarantees and presents experiments on 2D illustrative tasks and a molecular conformer generation task (QM9).

**Strengths:**

* Addresses how to leverage pre-trained generative models to explore novel and valid regions of a design space, moving beyond the original data distribution.
* Formalizes the problem into Global Flow Expansion (using a strong verifier) and Local Flow Expansion (using a weak verifier).
* Lifts the optimization objective from the final time-step ($p_1$) to the entire noised state space ($Q^\pi$) to theoretically mitigate the score divergence problem that occurs as $t \to 1$.

**Weaknesses:**

* One weakness is the use of potentially uninformative baselines. The paper compares FE (a "search + constraint" method) against "search-only" (S-MEME/FDC) and "constraint-only" (CONSTR). This comparison is not fully informative, as FE is designed to outperform them. A fair and important baseline would be unconstrained exploration (FDC/S-MEME) followed by post-hoc rejection sampling using the verifier. Without this, the practical value of FE's complex optimization is unknown.
* The method's reliance on a differentiable verifier is a considerable practical limitation. The chosen solver (Adjoint Matching, Alg. 3) requires gradients from $\log v(x)$. Most real-world verifiers for scientific discovery (e.g., RDKit SanitizeMol, physics simulators) are black-box and non-differentiable. The paper's workaround (smoothing a simple function in App G.1, G.5.2) does not solve this general problem, limiting the method's applicability.
* The experimental validation is not fully convincing. The 2D experiments can be sensitive to tuning, and the QM9 dataset is too small-scale to demonstrate robustness.
* There is a notable absence of hyperparameter ablation studies for the key parameters ($\alpha, \gamma_k, \eta_k$).
* This is particularly concerning for the L-FE molecular experiment. The chosen parameters ($\alpha=9$, $\gamma_k=0.00001/(1+k)$) are so conservative that the effective step size is $\tilde{\gamma}_k \approx 0.0001$ and the KL weight $\beta=0.9$. This suggests the model is moving very little from the pre-trained state. The claimed high validity (81%) is likely just the original model's high validity, not a product of the algorithm.
* As discussed in Appendix H, $\alpha$ and $\gamma_k$ are entangled, jointly determining the effective step size $\tilde{\gamma}_k$. This implies the method is likely quite sensitive to hyperparameter tuning, but this important aspect is not analyzed.
* The computational cost appears very high. The algorithm requires $\approx 2 \times K \times N$ full model fine-tuning runs (e.g., $2 \times 8 \times 4 = 64$ in the L-FE experiment). This may be impractical for large-scale problems, and the paper makes no analysis of this cost or scalability against other baselines.
* A potential theory-practice gap exists. The convergence guarantees (Thm 5.2) rely on assumptions (E.1, E.2) about the solver, but there is no proof or justification that the actual solver used (Adjoint Matching, Alg. 3) satisfies these assumptions.

**Questions:**

* Can you provide a direct comparison against the FDC + post-hoc rejection sampling baseline? This would be very helpful to demonstrate that your complex constrained optimization is superior to a simple filter.
* How do you propose to use FE with a truly black-box, non-differentiable verifier (e.g., a hard RDKit sanitization check)? This seems to be the most common and important use case.
* Please address the question that your L-FE experiment parameters ($\alpha=9$, $\gamma_k \approx 0.00001$) are so conservative the model is "not moving" from the pre-trained state. Can you provide a hyperparameter ablation study for $\alpha$ and $\gamma_k$ to show how the diversity/validity trade-off changes?
* To help us understand the method, can you provide an ablation showing the results of running only the EXPAND step and only the PROJECT step for $K$ iterations?
* Can you provide an experiment for Global Flow Expansion on a real-world dataset, not just a 2D toy problem?

I would be willing to raise my score if the authors can thoroughly address the concerns raised in the weaknesses and questions section. In particular, the concerns regarding the experiments (ablation, large-scale dataset).

---

> ### Author Response · Authors · 2025-11-25
>
> We thank the Reviewer for reporting detailed concerns regarding the experimental evaluation of the proposed methods. As the Reviewer can see within the updated version of our paper, we have performed a significant expansion of the experimental section using one extra page as allowed by ICLR guidelines, with the intent to tackle all points raised by the Reviewer. Within the Global response, we have presented the main new experimental contributions.
> For the sake of clarity, we first wish to mention to the Reviewer that within the updated version of the work, we have introduced a method for (unconstrained) exploration named Noised Space Exploration (see Alg. 3), which corresponds to projection-free FE. As explained within the Global response, NSE shows higher performance for unconstrained exploration than FDC, thus further clarifying the relevance of this work for the flow-based exploration literature, and disentangling the roles of better exploration capabilities and verifier use in the experimental evaluation of FE.
>
>
>
> Given these premises, in the following, we aim to sharply discuss each specific point raised by the Reviewer.
>
>
> **Limited experimental validation**
>
> Towards strengthening our experimental validation, we have evaluated our method on a de novo design task on GEOM-Drugs [5], which we believe to be significantly more informative than QM9 and closer to the complexity of real-world applications. This is presented within the new paragraph "L-FE increases molecular conformer diversity for De-Novo Design on GEOM-Drugs" in Sec. 6. In short, we show that L-FE shows superior performance, both in terms of induced diversity and validity, compared against a state-of-the-art flow-based exploration method (FDC [4], a NeurIPS 2025 Spotlight paper). We believe that this new experimental evaluation on GEOM-Drugs, a highly more complex and high-dimensional setting than QM9, further showcases promising performance of the proposed constrained exploration scheme in a setting closer to real-world discovery tasks, on which the application of L-FE would be a natural next step.
>
>
> **Differentiable vs non-differentiable verifier**
>
> None of the proposed algorithms (L-FE, G-FE) rely on a differentiable verifier for the verifier-based projection step. In particular, a 1-st order (i.e., gradient-based) fine-tuning scheme (e.g., Adjoint Matching) is only required to maximize entropy within the expansion step (line 2, Alg. 1). This is the case because the gradient of the entropy first variation admits a closed-form expression (see Eq. 15), while the first variation does not. Nonetheless, the projection step (line 3, Alg. 1) does not require to employ the same fine-tuning method, instead one can leverage any off-the-shelf 0-th order (i.e., gradient-free) fine-tuning scheme (e.g., RL-based [1,2], DPO [4] etc.) to enforce the verifier constraints. We thank the Reviewer for raising this important point, and have clarified it within the updated manuscript. Moreover, several recent works tackled the problem of constrained fine-tuning corresponding to our projection step [e.g., 4]. Crucially, these methods, as well as future ones, can be leveraged to implement the projection step within our constrained entropy-maximization scheme (Alg. 2) depending on the type of available verifier.
>
>
>
> **Hyperparameter ablation**
>
> We performed an ablation study on the aforementioned GEOM-Drugs setting, reported within Apx. G.5.4 of the updated paper. See Fig. 7 for ablations of the $\alpha$ (which we express via $\beta$ through the reparameterization presented in Appendix H), $\eta_k$, and $\gamma_k$ parameters.
>
>
> **High validity due to prior regularization rather than constraint enforcement**
>
> As described in Sec. 3.2, local flow expansion preserves (and potentially increases) validity both via prior regularization, and by leveraging the validity signal within the used verifier. As one can notice in the new Fig. 6.a (for GEOM-Drugs), L-FE can even achieve significantly higher validity compared to the pre-trained model (PRE), while vastly increasing the diversity metric. This observation seemingly implies that L-FE can properly incorporate validity information via the verifier-based projection step, rather than simply preserving validity via pre-trained model regularization.

---

> > ### Author Response · Authors · 2025-11-25
> >
> > **Inference time filtering baseline**
> >
> > We aim to tackle this point raised by the Reviewer via the following two observations.
> >
> > 1. We wish to point out the fact that the problem tackled within our work regards the adaptation at fine-tuning time of a pre-trained generative model, as shown in Fig. 1, by expanding beyond high-density regions while preserving sufficient validity. This is a fundamentally different problem than computing an expanded flow model and then enforcing validity at inference-time via filtering. In particular, while the former approach entails a one-time heavy computational process, the latter implies a (potentially) significantly higher per-sample sampling cost at inference-time due to the filtering process, which is clearly prohibitive for some use cases (e.g., virtual screening), where one wishes to generate a huge amount of molecules while potentially adhering to costly verifiers. Similarly, specific verifiers might be proprietary/private and not available at sampling time to the end user of the expanded generative model, or a company might wish to release a high-validity generative model not requiring per-sample correction/filtering. In conclusion, we believe that the method we propose is highly relevant for several realistic settings where the proposed baseline does not provide a reasonable alternative.
> >
> > 2. Nonetheless, for the sake of completeness, we report in Appendix G.5.7 an experimental comparison of the following methods: (a) FDC + inference-time filtering, (b) NSE + inference-time filtering, and (c) L-FE. Within Apx. G.5.7 we report a detailed discussion of the results. In short, NSE + inference-time filtering significantly outperforms FDC + inference-time filtering, while L-FE achieves superior exploration capabilities compared against FDC + inference-time filtering, while suffering an expectable minor validity drop. In particular, we wish to point out that these algorithms have theoretically the same closed-form solutions, and the concrete differences amount to (1) the high per-sample cost of inference-time filtering, and (2) the superior exploration capabilities of L-FE and NSE over FDC likely due to noised space exploration, as discussed in Sec. 4.
> >
> > In conclusion, if inference-time filtering (i.e., higher per-sample sampling cost) is permitted, then NSE (introduced in Alg. 3) shows improved capabilities compared with current methods, while L-FE is to our knowledge the first scheme that solves the fine-tuning counterpart of the problem, where one wishes to obtain a good quality model without employing per-sample correction/filtering schemes, as discussed within point (1) above. In both cases, the algorithms presented within this work (i.e., NSE and L-FE) dominate existing methods.
> >
> >
> > **Entangled hyperparameters**
> >
> > The discussion in Appendix H entitled "Update step reparametrization" is intended to address that exact point. In particular, we tuned the hyperparameters of our method by using such reparametrization (i.e., via $\beta$ ) so that the parameters are significantly less entangled and therefore the search process for finding good parameterization becomes significantly easier. We present the original parametrization in the main paper as it is easier to associate to standard mirror descent schemes and our theoretical results.
> >
> > **Computational Cost**
> >
> > We performed a computational cost (runtime) comparison between our methods (L-FE, and the newly introduced NSE) and FDC on GEOM-Drugs. We reported the results within Fig. 6.c and discussed them within the paragraph "L-FE and NSE have computational costs comparable to current exploration schemes.", which title already expresses the main conclusion of this experimental evaluation. In particular, the computational overhead required by FE and NSE to perform noised space exploration seems marginal compared to the total computational cost of current schemes, while it allows to achieve stronger performance in higher-dimensional settings (see e.g., Fig. 6.b for a unconstrained exploration comparison of NSE vs FDC).

---

> > > ### Author Response · Authors · 2025-11-25
> > >
> > > **Theory-practice gap**
> > >
> > > We thank the reviewer for pointing out the potential theory–practice gap in Theorem 5.2. In the following, we aim to further clarify these points.
> > >
> > > 1.  (E.1) (precompactness of the iterates). In our finite-dimensional parameter space, (E.1) essentially requires that the sequence of iterates produced by the solver remains in a bounded set. This is a very mild requirement: it is satisfied as soon as the solver does not diverge numerically (e.g., no exploding parameters or NaNs), which is exactly what we observe in all our experiments. Moreover, standard practices such as bounded initialization, weight decay, and gradient clipping can be viewed as explicit mechanisms that enforce this boundedness. We clarified this point and the interpretation of (E.1) in the revised version.
> > >
> > > 2. (E.2) (solver accuracy). Assumption (E.2) formalizes that the inner problem is solved to a certain accuracy. For our nonconvex deep models, establishing worst-case guarantees that a particular optimizer (Adjoint Matching / SGD-based solver) always attains this accuracy is out of reach in general, since globally optimizing such models is known to be NP-hard. As a result, our theorem is intentionally conditional on (E.2), in exactly the same spirit as many theoretical results in deep learning and bilevel optimization that assume access to a sufficiently accurate inner solution without proving that a specific training heuristic will always find it. In practice, it is possible to monitor the inner objective and run the solver until convergence plateaus; empirically, the method behaves robustly and achieves strong performance, which indicates that the required accuracy is met in the regimes we study. We will make this relation to existing deep-learning theory more explicit in the paper.
> > >
> > >
> > >
> > > **Ablation for only expansion and only projection**
> > >
> > > Within the updated version of the work, we report in Appendix G.5.5 the results of running only the EXPAND steps and only PROJECT steps (Fig. 8). Crucially, notice that running only EXPAND steps corresponds to NSE, i.e., the newly introduced algorithm (see Alg. 3) for unconstrained expansion. As one can notice in Fig. 8, sequential EXPAND steps gradually decrease validity and increase diversity. Analogously, sequential projection performs approximately as an idempotent map, preserving or slightly decreasing diversity, and slightly increasing validity, due to its approximate nature.
> > >
> > >
> > > **G-FE on real-world dataset**
> > >
> > > We report G-FE run for $K=3$ on the GEOM-Drugs dataset within Fig. 7 in Appendix G.5.4. We observe that it shows competitive performance, slightly dominated in terms of validity by L-FE, likely due to stronger prior regularization. Nonetheless, we wish to point out that for any finite number $K$ of iterations, G-FE can effectively approximate the asymptotic solution of L-FE, due to KL-bounded iterations between mirror descent iterates. In particular, we introduced the distinction between L-FE and G-FE for enhanced controllability via the extra KL term of L-FE (see Eq. 9), and to be able to guarantee asymptotically optimal solutions for respectively local and global expansion and therefore close a potential theory-practice gap.
> > >
> > >
> > > We hope that the presented new contributions and responses could tackle the Reviewer's concerns. If this is not the case, we are happy to provide further clarification.
> > >
> > > **References**
> > >
> > > [1] Training diffusion models with reinforcement learning, K. Black, 2023.
> > >
> > > [2] Dpok: Reinforcement learning for fine-tuning text-to-image diffusion models, Y. Fan, 2023.
> > >
> > > [3] Diffusion model alignment using direct preference optimization, B. Wallace, 2024.
> > >
> > > [4] Constrained Molecular Generation via Sequential Flow Model Fine-Tuning
> > >
> > > [5] GEOM, energy-annotated molecular conformations for property prediction and molecular generation, Simon Axelrod and Rafael Gomez-Bombarelli, 2022.

---

### Official Review · Reviewer_oQPY · 2025-10-31

**Soundness:** 4
**Presentation:** 3
**Contribution:** 3
**Rating:** 8
**Confidence:** 3

**Summary:**

Flow and diffusion models are typically pre-trained on limited available data. As a result, they tend to generate samples from only a narrow portion of the feasible domain.
To address this limitation, the authors assume access to a verifier and propose adapting a pre-trained flow model so that its induced density expands beyond regions of high data availability. They pose the key question:
“How can we leverage a given verifier to adapt a flow or diffusion model to generate designs beyond high data-availability regions while preserving validity?”
The authors consider two types of verifiers:
•	Strong verifier: a function nu: X -> {0,1} that characterizes validity exactly, i.e., nu(x)=1 if and only if x is valid.
•	Weak verifier: a function that acts as a filter—it rejects some invalid designs but may fail to detect others (formally, nu(x)=0 => x is invalid).

**Strengths:**

Major contributions:
•	Flow Expander (FE), a principled probability-space optimization scheme
•	A theoretical analysis of the proposed algorithm
•	An experimental evaluation of FE

It is a well-written paper, with new ideas and interesting results.

I think many researchers in our community will appreciate this paper.

**Weaknesses:**

•	The paper is somewhat dense and not always easy to follow.

•	The numerical experiments are somewhat limited. I appreciate both the illustrative examples and the results on the molecular design task, but I wish the paper included more high-impact, real-world examples where verifiers exist.

**Questions:**

•	In the description of Continuous-time Reinforcement Learning, states and actions have been defined, but I think the definition of reward is missing.

•	Line 175: If Omega_v is not a bounded set, then without further constraints, there is no maximum entropy distribution. Is there a simple way to generalize Problem 5 to this unbounded setting?

•	Instead of maximizing entropy, can we get reasonably good results by simply maximizing the variance of the distribution instead?

---

> ### Author Response · Authors · 2025-11-25
>
> We thank the Reviewer for finding our work as well-written, with new ideas and interesting results, and ultimately thinking that 'many researchers in our community will appreciate this paper'. In the following, we sharply address the limitations and questions mentioned within the review.
>
> **Paper writing is somewhat dense**
>
> We thank the Reviewer for raising this point. Unfortunately this was mostly due to space limits. We aimed to relax the writing, but this proved complicated due to the new extra page of experimental results. Depending on the final length of the work, we will make sure to maximize clarity by de-compressing the writing, especially in the more technical parts, in the updated version.
>
> **Limited experiments**
>
> This work primarily aims to provide a mathematical framework and theory-backed algorithmic scheme to tackle the highly relevant problem of verifier-constrained flow expansion. While we believe that the nature of the work is primarily algorithmic, we agree with the Reviewer regarding the limitations of the experimental evaluation provided within the previous version of the paper. As a consequence, within the updated version, we have significantly extended the experimental section using the last extra page as by ICLR guidelines. We included multiple experimental analyses aiming to further close the gap between the newly introduced algorithmic scheme and real-world applications. To this end, we have have run our method on a de novo design task on GEOM-Drugs [1], which we believe to be significantly more indicative than QM9 and closer to the complexity of real-world applications. This new application is presented within the new paragraph "L-FE increases molecular conformer diversity for De-Novo Design on GEOM-Drugs" in Sec. 6. In short, we show that L-FE shows superior performance, both in terms of induced diversity and validity, compared against a state-of-the-art flow-based exploration method (FDC [4], a NeurIPS 2025 Spotlight paper). We believe that this new experimental evaluation on GEOM-Drugs, a highly more complex and high-dimensional setting than QM9, further showcases promising performance of the proposed constrained exploration scheme in a setting closer to real-world discovery tasks, on which the application of FE would be a natural next step. Moreover, while we present applications only within the field of scientific discovery, rather than image generation etc., we believe that such settings (i.e., molecular/drug design) are highly relevant for high-impact real-world applications (e.g., drug discovery).
>
> **Questions**
>
> 1. Within the context of this work, reward functions over $\mathcal{X}$ are defined by $f_t(x)$, as reported within Sec. 4.
>
> 2. We thank the Reviewer for posing this interesting question! We wish to point out that, from a certain standpoint, the Local Expansion formulation in Problem 6 can effectively answer the question, as the KL regularization can be interpreted via a Lagrangian viewpoint as an effective constraint, thus rendering the optimization problem to not require boundedness anymore. Intuitively, this would correspond to artificially bounding the generative models search space to an hyper-ball in probability space centered around the pre-trained model $\pi^{pre}$. Different values of regularization strength $\alpha$ would then lead to more or less conservative effective constraints (i.e., hyper-ball radius). Moreover, one could extend the presented formulation to impose a divergence constraint of the form $D(p^\pi, p^{pre}) \leq B$ to explicitly control the expansion process over originally unbounded domains.
>
> 3. While the entropy functional leads to a closed-form update rule for optimization schemes (see Eq. 15), this is (in general) not the case for the variance. As a consequence, it is not obvious how one could control the distribution's variance. Moreover, while entropy-based exploration is strongly motivated by the reinforcement learning literature (e.g., [1,2]), this is arguably not the case for variance maximization.
>
>
>
> **References**
>
> [1] The Importance of Non-Markovianity in Maximum State Entropy Exploration, Mirco Mutti et al., 2022.
>
> [2] Behavior from the void: Unsupervised active pre-training, Hao Liu et al., 2021.

---

### Official Review · Reviewer_Kx6q · 2025-11-01

**Soundness:** 3
**Presentation:** 2
**Contribution:** 2
**Rating:** 4
**Confidence:** 1

**Summary:**

This paper proposes Flow Expander (FE), a framework that expands the support of pretrained flow or diffusion models under verifier constraints. The method formulates verifier-constrained entropy maximization as a mirror-descent optimization problem over probability measures, consisting of two alternating steps: an Expansion step that increases entropy and explores new modes, and a Projection step that enforces validity through a soft-verifier function. The authors provide theoretical convergence guarantees and demonstrate the approach on toy 2D examples and QM9 molecular conformers. The work is theoretically well-motivated, connecting flow-based generative modeling with constrained optimization, but its empirical validation and baseline clarity remain limited.

**Strengths:**

**Strong motivation and relevance.**
The paper tackles an important limitation of pretrained flow and diffusion models—namely, their inability to explore beyond the data manifold while maintaining sample validity. The idea of integrating verifier-based constraints into generative model fine-tuning is timely and relevant for scientific design applications (e.g., molecular or material generation).

**Principled formulation.**
The proposed *Flow Expander* framework is grounded in a clear optimization principle: verifier-constrained entropy maximization. Casting this as a mirror-descent problem in the space of probability measures provides a mathematically elegant and unified view of exploration under validity constraints.

**Theoretical completeness.**
The paper presents a solid convergence analysis. The proofs, though dense, follow established mirror-descent theory and offer formal justification for the proposed update rule.

**Potentially general concept.**
The introduction of a *soft-verifier* mechanism is conceptually interesting and, if properly extended, could provide a practical interface between learned generative models and rule-based or simulation-based validity filters used in real-world design workflows (e.g., high-throughput screening, physical constraints, or chemical property checks).

**Weaknesses:**

**Limited and unconvincing experiments.**
The empirical evaluation is restricted to 2D toy examples and a small-scale QM9 conformational generation task. These setups are insufficient to demonstrate practical effectiveness or scalability of the proposed method. The results primarily serve as proof-of-concept demonstrations rather than evidence of real-world impact or generalization capability.

**Unclear and unverifiable baselines.**
The paper cites Uehara et al. (2024, Section 8.2) as the source of the “CONSTR” baseline. However, that section contains only a theoretical corollary without any algorithmic description, implementation details, or experimental setup, making it impossible to reproduce or verify the reported baseline results. Moreover, many components of the proposed method appear to be directly borrowed from S-MEME (e.g., the mirror-descent formulation, entropy maximization objective, and theoretical convergence argument), yet the paper does not clearly delineate which parts are newly introduced and which are adapted from prior work. This lack of transparency in baseline selection and methodological novelty significantly undermines the credibility and reproducibility of the experimental claims.

**Severe clarity and notation issues.**
The paper suffers from significant readability and presentation problems:
1. Misuse of notation and inconsistent subscripts/superscripts.
2. line (234) incorrectly states $\delta\mathcal{G}(\mu)\in F(\mathcal{X})$, though the functional derivative should be a function over $\mathcal{X}$.
3. Algorithm 4 is referenced but missing.
4. Symbols such as $F(\mathcal{X}), \mathcal{G}_t$, and $\mathcal{L}(Q)$ are repeatedly overloaded, making the derivations unnecessarily hard to follow. These issues significantly reduce the accessibility of an otherwise theoretically interesting paper.

**Applicability and experimental realism.**
The introduction of *soft-verifiers* is an interesting and potentially powerful concept, as it could, in principle, allow high-throughput filtering strategies—commonly used in drug and material discovery—to be incorporated into generative models. However, the paper does not provide convincing examples where this mechanism becomes practically important. Beyond toy examples, the molecular conformational experiments hold limited chemical relevance: in conformational space, the goal is typically to reproduce the Boltzmann distribution rather than to impose external validity filters, so the verifier concept is of marginal utility.

**Questions:**

**Separation from S-MEME.**
Many components of the proposed Flow Expander (e.g., entropy maximization, mirror-descent formulation, convergence proof) appear closely aligned with S-MEME. Could the authors clearly specify which elements are newly introduced in this paper (e.g., verifier projection) and which are inherited from S-MEME or related prior works?

**Applicability to discrete domains.**
The proposed formulation assumes a continuous variable space, where functional gradients and mirror-descent updates are well defined.
How could this framework extend to **discrete or combinatorial domains** such as molecular graphs or protein sequences, where the notion of a variational gradient is not clearly defined?
In particular, can the *soft-verifier* idea be adapted to these settings, which are arguably more relevant to real-world drug or material discovery?

**Details Of Ethics Concerns:**

The reviewer utilized a large language model (LLM) to assist in reviewing and interpreting the heavy mathematical parts of the paper.

---

> ### Author Response · Authors · 2025-11-25
>
> We thank the Reviewer for recognizing that our work tackles a problem with strong motivation and relevance, the idea of integrating verifier-based constraints as timely and potentially general, the formulation as mathematically elegant and unifying, and the  presented theory as solid. In the following, we sharply address several limitations and questions mentioned within the review.
>
> **Limited and unconvincing experiments**
>
> As mentioned by the Reviewer, this works provides a 'mathematically elegant and unified view of exploration under validity constraints' and a theory-backed algorithmic scheme to tackle this highly-relevant problem. While we believe that the nature of the work is primarily algorithmic, and that real-world evaluation of the proposed method goes beyond the scope of this work, we agree with the Reviewer regarding the limitations of the experimental evaluation provided within the previous version of the paper. As a consequence, within the updated version, we have significantly extended the experimental section by using one extra page as by ICLR guidelines. We included multiple experimental analyses aiming to further close the gap between the newly introduced algorithmic scheme and real-world applications. To this end, we have have run our method on a de novo design task on GEOM-Drugs [3], which we believe to be significantly more indicative than QM9 and closer to the complexity of real-world applications. This is presented within the new paragraph "L-FE increases molecular conformer diversity for De-Novo Design on GEOM-Drugs" in Sec. 6. In short, we show that L-FE shows superior performance, both in terms of induced diversity and validity, compared against a state-of-the-art flow-based exploration method (FDC [4], a NeurIPS 2025 Spotlight paper). We believe that this new experimental evaluation on GEOM-Drugs, a highly more complex and high-dimensional setting than QM9, further showcases promising performance of the proposed constrained exploration scheme in a setting closer to real-world discovery tasks, on which the application of L-FE would be a natural next step.
>
> **Baselines and relation to SMEME/FDC**
>
> 1. Concretely, the CONSTR baseline corresponds to the projection step as employed within Alg. 1 (line 3), with pseudocode provided in Alg. 4 within Appendix F. This method is effectively an implementation of the constrained generation scheme proposed by Uehara et al. (2024, Section 8.2). We thank the Reviewer for mentioning this point, and we updated the uploaded version of the paper to clarify it.
>
> 2. The fundamental differences with S-MEME and FDC are two-folds:
>
> - (a) we consider a verifier-constrained setting, while S-MEME and FDC tackle an unconstrained exploration problem. Besides the problem formulation, this difference also leads to a substantially different algorithmic scheme (i.e., Flow Expansion), which relies on the ExpandThenProject sub-routine (Alg. 1), that alternates expansion steps with projection steps to ensure constraints satisfactions.
>
> - (b) S-MEME and FDC rely only on the score information at time-step $t=1$, while FE effectively performs entropy maximization over the entire noised space, effectively optimizing the integral of an entropy functional over the entire flow process path (see Eq. 8 in Sec. 4). Algorithmically, this corresponds to a mirror descent scheme over the probability space of entire flow processes (i.e., all-times marginal densities) rather than only on the space of last-time-step marginal densities. As mentioned within Sec. 4 (lines 314-320), this fundamental algorithmic difference renders possible to obtain (as a by product) also a more stable algorithm for unconstrained exploration. To further clarify this point, within the updated version of the work we have explicitly introduced such a new method by the name of Noised Space Exploration (NSE) as an independent contribution (see the new Alg. 3 in Sec. 4). Crucially, within the updated Sec. 6, we perform an experimental evaluation of NSE showing significant performance improvement over FDC for unconstrained exploration tasks both in terms of diversity and validity. This is presented within the new paragraph "NSE achieves higher exploration performance against current methods." and in Fig. 6.b within Sec. 6. We thank the Reviewer for raising this point, and believe that the updated version of the paper sheds light on the independent contribution/relevance of NSE, which corresponds to FE without projection step, w.r.t. the recent (unconstrained) diffusion/flow-based exploration literature. Concretely, this clarification shows that NSE outperforms a state-of-the-art method for (unconstrained) flow-based design space exploration.
>
> Moreover,  the theoretical analysis significantly extends prior works by (1) accounting for the distributional constraint  $\pi : p_0^\pi = p_0^{pre}$ (see Eqs. 5 and 7) and (2) analyzing optimization over the entire flow process rather than the last marginal density.

---

> > ### Author Response · Authors · 2025-11-25
> >
> > **Notational issues**
> >
> > - The first variation $\delta \mathcal{G}(\mu)$ can be interpreted as a linear functional over $\mathcal{X}$, rendering the current notation correct. Nonetheless, we agree with the Reviewer that the function viewpoint is easier to parse. As a consequence, we have updated this within the paper, and we thank the Reviewer for suggesting this presentational change.
> >
> > - The pointer to Alg. 4 should actually refer to Alg. 1. We thank the reviewer for noticing this typo, which we fixed in the updated manuscript.
> >
> > - The symbols $F(\mathcal{X}), \mathcal{G}_t$ and $\mathcal{L}(Q)$ are used to denote fundamentally different objects. In particular, the first refers to a general functional over the fixed probability space, the second (i.e., $\mathcal{G}_t$) to the objective defined within Eq. 9, while the third (i.e., $\mathcal{L}(Q)$) refers to the integral of $\mathcal{G}_t$ along the time dimension of the flow process, as shown in Eq. 8. We are not sure about what the Reviewer concretely wished to express by this comment. Please, let us know in case we did not understand.
> >
> >
> > **Experimental realism**
> >
> > We believe there might be some confusion regarding the chemical application modeled within Sec. 6. The presented molecular design application is in the context of de novo molecular design, where the pre-trained generative model models entire molecular geometric graphs. We do not tackle the classic problem of graph conditioned conformer generation as mentioned by the Reviewer. Withing the context of de novo design, we employ FE to increase the diversity of the generated conformers (i.e., the 3D structure of generated molecules). Since this case study models a de novo design task, we believe that it already represents a relatable setting where verifier knowledge is indeed useful in real-world applications to enforce synthesizability, or any other constraint on the generated molecules, while inducing high diversity.
> >
> > **Applicability in discrete domains**
> >
> > The presented constrained entropy-maximization framework can indeed be leveraged for discrete or combinatorial domains. In particular, the variational gradient of the entropy functional, namely $\delta \mathcal{G}_t$ can be identically defined for discrete spaces (e.g. proteins) as presented in Sec. 4. On the other hand, discrete domains do not admit a typical gradient (w.r.t. $x$), and therefore it would be not obvious to compute an object analogous to $\nabla_x \delta \mathcal{G}_t$, as used within line 4 of Alg. 2. But crucially, this is not required to implement the algorithm in discrete spaces, as it is sufficient to define the surrogate reward functions $f_t = \delta \mathcal{G}_t$ and use any off-the-shelf RL-based fine-tuning method, which does not require gradient information. Importantly, $f_t = \delta \mathcal{G}_t$ can be straightforwardly expressed for LLMs via log density computation, and similarly it can be (approximately) estimated for certain types of discrete diffusion models, e.g., via [1, Alg. 3], which have recently found significant applications in real-world problems such as enzyme design (see e.g., [2]). In conclusion, nearly all contributions in this work extend naturally to discrete or combinatorial domains, as asked by the Reviewer.
> >
> >
> > We hope that the presented new contributions and responses could tackle the Reviewer's concerns. If this is not the case, we are happy to provide further clarification.
> >
> > **References**
> >
> > [1] Large Language Diffusion Models, Shen Nie et al., 2025
> >
> > [2] Steering Generative Models with Experimental Data for Protein Fitness Optimization, Jason Wang, 2025.
> >
> > [3] GEOM, energy-annotated molecular conformations for property prediction and molecular generation, Simon Axelrod and Rafael Gomez-Bombarelli, 2022.
> >
> > [4] Flow Density Control: Generative Optimization Beyond Entropy-Regularized Fine-Tuning, Riccardo De Santi et al., 2025.

---

### Author Response · Authors · 2025-11-25
**Global Response**

We thank the Reviewers for recognizing our paper as well-written (Reviewer oQPY), the problem treated as relevant and with strong motivation (Reviewer Kx6q), the formulation principled, mathematically elegant, and unifying (Reviewer Kx6q), the presented theory solid (Reviewer Kx6q), the method practical (Reviewer e76w), and for stating that "many researchers in our community will appreciate this paper". We are very grateful for such positive comments as well as for the constructive feedback provided by the Reviewers. We have significantly improved the paper by using one extra page (see blue text within the updated version), according to ICLR guidelines, aiming to address all Reviewers concerns. Among other things, within the updated version of the manuscript, we have vastly expanded and refined the evaluation of the proposed method via the following new contributions:

**Experimental evaluation on GEOM-Drugs**

Towards tackling the concerns regarding the limitations of the experimental evaluation on QM9, we have expanded the experimental evaluation of FE on the GEOM-Drugs dataset [1], which we believe to be significantly more indicative than QM9 and closer to the complexity of real-world applications. This is presented within the new paragraph "L-FE increases molecular conformer diversity for De-Novo Design on GEOM-Drugs" in Sec. 6. In short, we show that L-FE shows superior performance, both in terms of induced diversity and validity, compared against a state-of-the-art flow-based exploration method (FDC [2], a NeurIPS 2025 Spotlight paper).

**Evaluation for unconstrained exploration against current methods**

From an algorithmic standpoint, Flow Expansion (FE) introduces innovations with the two-fold objective of:

1. Improving the exploration capabilities of existing diffusion/flow-based exploration schemes, by stabilizing the exploration process via lifting it to the noised state space (see Sec. 4, lines 307-325 for details), and

2. Incorporating verifier information within the exploration process, effectively being, to our knowledge, the first verifier-based expansion method.

While Reviewers acknowledged the clarity and novelty regarding point (2), several Reviewers rightfully pointed out the lack of clarity within the previous version of our work regarding point (1), namely the capabilities of unconstrained exploration of the proposed method (FE) compared with prior schemes for (unconstrained) diffusion-based exploration. Towards clarifying this point, within the updated version of the work we introduced a new method, Noised Space Exploration (NSE), see Alg. 3 within Sec. 4, which corresponds to a projection-free version of FE. This renders possible to properly assess the contributions of this work for the (unconstrained) exploration problem and literature. Crucially, we added a new paragraph "NSE achieves higher exploration performance against current methods." in Sec. 6, where we experimentally evaluate NSE, showing superior performance against FDC [2, NeurIPS 2025 Spotlight] (which is strictly more general than S-MEME [3]).

**FE ablation and computational cost analysis**

1. We conduct an ablation study for the proposed method on GEOM-Drugs [1] (see Appendix G.5.4), showing detailed comparison of L-FE, G-FE, NSE, and FDC.

2. We included an experimental evaluation of the computational cost of the proposed methods compared with current exploration schemes, within the new paragraph "L-FE and NSE have computational costs comparable to current exploration schemes." in Sec. 6.

We believe that the updated experimental evaluation and introduction of the (unconstrained) exploration scheme (i.e., NSE) render the updated work significantly more convincing than its previous version. In particular, while the presented experiments do not directly show real-world impact of the presented method, they showcase that it is highly promising and of high relevance for this problem and literature. Since this work is primarily of algorithmic nature, rather than applied, we believe that the presented contributions are strong, and hope the Reviewers can further appreciate them via the updated version of the work, which aimed to settle all concerns raised by the Reviewers.


**References**

[1] GEOM, energy-annotated molecular conformations for property prediction and molecular generation, Simon Axelrod and Rafael Gomez-Bombarelli, 2022.

[2] Flow Density Control: Generative Optimization Beyond Entropy-Regularized Fine-Tuning, Riccardo De Santi et al., 2025.

[3] Provable Maximum Entropy Manifold Exploration via Diffusion Models, Riccardo De Santi et al., 2025.

---

### Meta-Review · Area_Chair_i1cv · 2025-12-29

**Summary:**

This paper tackles the important problem of helping generative flow matching models reliably extrapolate beyond their training data. This is especially important in domains where the training data does not “densely” cover the space of feasible designs, or where the feasible space is only partially represented. This is typical in domains such as small molecules (drug design), proteins, materials etc. The authors propose to incorporate “verifiers” (strong and weak) into a flow generation process that also incorporates an entropic reward function (encouraging exploration) for generating designs beyond the support of the pre-trained model, but while also respecting design validity constraints. The proposed method, Flow-Expansion (FE) uses a mirror descent style algorithm over the entire flow process, it is theoretically analyzed, then subject to a number of toy and realistic generation tasks.


Strengths:

- Tackles a relevant and important task constraining the use of pretrained generative models for domains that do not have internet-scale training data.
- Principled formulation (optimization and projection over the whole flow process, local and global formulations, weak and strong verification etc.) and thorough analysis justifying algorithmic design choices.
- The above are a major contribution.
- For the most part well written and nicely formatted

Weaknesses:

- Initially limited empirical evaluation
- Clear delineation of contribution over existing works (e.g. S-MEME) and theory-practice gap.
- Not the easiest paper to follow, perhaps some work could be done to clarify the exposition.
- Baselines are search/exploration only or constraint only, not a combination (perhaps a simple constraint step could be added to the exploration only methods).
- Most useful verifiers are non-differentiable, and it is not obvious how to incorporate these.
- Hyperparameter setting advice and ablations missing

The authors do a stellar job of addressing the reviewers concerns by adding experiments, ablations and a new method (NSE) to help to understand their contribution. This paper was already methodologically interesting and strong, and now it also has the empirical validation to back its claims. I suggest this paper for acceptance into ICLR with a few minor changes to the camera ready version.

The presentation of the method should be simplified/clarified where possible as a number of the reviewers struggled with this. Also there may be some slight issues with references, e.g. in the appendix a ref to Alg. 8 is given, but it does not exist, some of the equation numbering is a little unintuitive (as equations are numbered in floating environments, making them out-of order. Perhaps consider giving a different numbering scheme to algorithm equations). Another work that is relevant and probably should be cited:

Klarner, L., Rudner, T. G., Morris, G. M., Deane, C., & Teh, Y. W. (2024, July). Context-Guided Diffusion for Out-of-Distribution Molecular and Protein Design. In International Conference on Machine Learning (pp. 24770-24807). PMLR.

**Reviewer Concerns:**

Concerns addressed:

- Limited empirical evaluation — GEOM-Drugs benchmark is added and L-FE is compared favourably to the SOTA FDC method.
- Contribution over related methods (S-MEME and FDC) — clarified in discussions and paper, also NSE introduced (L-FE without constraint projection for unconstrained generation), which is also shown to improve over baselines and to aid understanding of how entropy maximisation over whole flow process contributes to performance. Theory-practice gap concerns are addressed in the discussion with MbXc.
- Reward (entropy) gradient can be estimated in discrete domains using flow or diffusion models exactly by eqn. 19 or the score function respectively.
- Uninteresting baselines: an additional experiment is given in A G 5.7 showing the effects of inference-time filtering (on NSE and FDC) compared to projection (L-FE), and a discussion is given on the requirements and trade-offs of training and inference time verification constraint enforcement. This is an interesting an nuanced point, and greatly increases the applicability of the work.
- Computational cost concerns addressed with additional evaluations given in Fig 6c.
- A substantial number of hyperparameter ablations are added to the appendix, as well as detail for the experiments and discussions on their coupling/sensitivity.
- Clarification that verifiers do not need to be differentiable, only the reward (entropy) term, which has been shown to have a computable form in the discrete setting with flow matching and diffusion.


Concerns unaddressed:

- Paper’s presentation could possibly be further clarified

**Reviewer Scores:**

Kx6q: 4 -> 6 at least. All concerns were addressed.

oQPY: 8 -> 8. This reviewer was already scoring the paper well, and their concerns were addressed.

MbXc: 2 -> 4 or 6 as their concerns have been largely addressed (as far as I can tell).

e76w: 6 -> 6 or 8 as this reviewer’s concerns did not seem major, and the authors gave thorough responses to the reviewer’s questions.

This would bring the paper's score somewhere in the range 6-7.

---

### Decision · Program_Chairs · 2026-01-26

Accept (Poster)